# Benchmarking the vertically integrated ice-sheet model IMAU-ICE (version 2.0)

Constantijn J. Berends[1], Heiko Goelzer[2], Thomas J. Reerink[3], Lennert B. Stap[1], Roderik S. W. van de Wal[1,4]

[1] Institute for Marine and Atmospheric research Utrecht, Utrecht University, Utrecht, The Netherlands
[2] NORCE Norwegian Research Centre, Bjerknes Centre for Climate Research, Bergen, Norway
[3] Royal Netherlands Meteorological Institute KNMI, De Bilt, The Netherlands
[4] Faculty of Geosciences, Department of Physical Geography, Utrecht University, Utrecht, The Netherlands

*Correspondence to*: Constantijn J. Berends (c.j.berends@uu.nl)

**Abstract.** Ice-dynamical processes constitute a large uncertainty in future projections of sea-level rise caused by anthropogenic climate change. Improving our understanding of these processes requires ice-sheet models that perform well at simulating both past and future ice-sheet evolution. Here, we present version 2.0 of the ice-sheet model IMAU-ICE, which uses the depth-integrated viscosity approximation (DIVA) to solve the stress balance. We evaluate its performance in a range of benchmark experiments, including simple analytical solutions, as well as both schematic and realistic model intercomparison exercises. IMAU-ICE has adopted recent developments in the numerical treatment of englacial stress and sub-shelf melt near the grounding-line, which result in good performance in experiments concerning grounding-line migration (MISMIP, MISMIP+) and buttressing (ABUMIP). This makes it a model that is robust, versatile, and user-friendly, which will provide a firm basis for (palaeo-)glaciological research in the coming years.

## 1 Introduction

Large-scale ice-sheet retreat is one of the most troubling long-term consequences of anthropogenic climate change (Oppenheimer et al., 2019; Fox-Kemper et al., 2021). Large uncertainties exist in projections of future ice-sheet retreat in strong warming scenarios, caused to a large extent by uncertainties in the dynamical response of the Greenland and Antarctic ice sheets (e.g. van de Wal et al., 2019). Since part of this response happens on centennial to millennial timescales, observational evidence alone cannot sufficiently reduce these uncertainties. Instead, models of large-scale ice-sheet dynamics should also be based on evidence of past changes in ice-sheet geometry.

Palaeoglaciological applications put different demands and constraints on ice-sheet models than future projections. Whereas future projections typically run for a few hundred years into the future at most (e.g. Goelzer et al., 2020b; Levermann et al., 2020; Seroussi et al., 2020; Sun et al., 2020), palaeo-simulations can cover periods of hundreds of thousands of years (e.g. Abe-Ouchi et al., 2013; Berends et al., 2018, 2019, 2021a; de Boer et al., 2013; Willeit et al., 2019), requiring a high

computational efficiency. Many physical processes that can be neglected on the relative short timescales of typical future projections, such as glacial-isostatic adjustment (GIA; e.g. de Boer et al., 2014), feedbacks of ice-sheet geometry on the regional climate (e.g. Berends et al., 2018, 2019), and changes in orbital configuration (e.g. Abe-Ouchi et al., 2013), need to be accounted for when the goal is to investigate multimillennial ice-sheet evolution. This means it is important that palaeo-

ice-sheet models are robust and flexible enough to allow for such processes to be included, excluded, or altered without too much effort by the user. In such applications, the large uncertainties in the forcing (e.g. palaeoclimate and palaeogeography), combined with the need for long simulations and consequently a high computational efficiency, lead to a different tolerance for errors in the ice dynamics than what is deemed acceptable in future projections.

In the early 2000's, the Institute for Marine and Atmospheric Research Utrecht (IMAU) developed ANICE, an ice-sheet-shelf model using hybrid SIA/SSA ice dynamics and a 3D thermodynamical module (Bintanja and van de Wal, 2008). Over the years this model has been continuously developed; a set-up was created where four copies of the model were coupled together to simulate the four large Pleistocene ice-sheets (North America, Eurasia, Greenland, and Antarctica) simultaneously (ANICE4; de Boer et al., 2013). Later, this four-region set-up was coupled to the global sea-level equation solver SELEN

(ANICE4-SELEN; de Boer et al., 2014). An inverse method of forcing the model using benthic $\delta^{18}O$ records was developed (de Boer et al., 2014), and a matrix method to force the model with GCM time slices was created (ANICE2.1; Berends et al., 2018) and later combined with the inverse forcing method to create a strategy to reconstruct past changes in $CO_2$ (Berends et al., 2019). Around 2016, development of IMAU-ICE started: the spiritual successor of ANICE, still solving the same physical equations, but with a thoroughly overhauled code structure, aimed at a wider range of possible applications, including future

projections. The first result of this effort, IMAU-ICE v1.0, has been used in several research projects involving future projections (e.g. Goelzer et al., 2018, 2020a, 2020b; Seroussi et al., 2019, 2020; Levermann et al., 2020; Sun et al., 2020; Edwards et al., 2021; Payne et al., 2021), as well as palaeo-applications (Bradley et al., 2018).

One particular aspect of the model that did not change much between subsequent members of the ANICE/IMAU-ICE model

family is the ice-dynamical solver, which was based on the hybrid SIA/SSA developed for PISM (Bueler and Brown, 2009). Since this heuristic method of combining the velocities from the two different approximations was first presented, its relative simplicity, computational efficiency, and good performance at simulating large-scale ice-sheet dynamics have led many research groups to adopt it as the basis for their own ice-sheet models (e.g. SICOPOLIS; Greve et al., 2011; f.ETISh; Pattyn, 2017; GRISLI; Quiquet et al., 2018; Yelmo; Robinson et al., 2020; UFEMISM; Berends et al., 2021b). However, the hybrid

SIA/SSA method has been shown to yield unsatisfactory results for geometries where features of the underlying bedrock are no longer very large or very small compared to the ice thickness (Goldberg, 2011). These shortcomings are considered to be small when the model resolution is large (as is the case for most palaeo-ice-sheet models, which typically use resolutions ranging between 10 – 100 km). However, using such a coarse resolution can result in the smoothing out of topographical

features such as fjords and pinning points, which can lead to erroneous ice velocities even when purely numerical errors are still small (Cuzzone et al., 2019).

Only two years after the popular hybrid SIA/SSA approach was published by Bueler and Brown (2009), another vertically integrated approximation to the stress balance was derived by Goldberg (2011). The "depth-integrated viscosity approximation" (DIVA) approximates the first-order momentum balance, and was derived using variational methods. It captures the longitudinal shear stresses included in the SSA, the vertical shear stress included in the SIA, plus the stress due to longitudinal stretching caused by vertical shearing (present in neither the SIA nor the SSA). It can therefore be viewed as a more mathematically consistent derivation of the dynamical equations which are heuristically approximated by the hybrid SIA/SSA. Goldberg (2011) applied the DIVA to the experiments from the ISMIP-HOM intercomparison (Pattyn et al., 2008), and showed that it produces results that agree with those from higher-order and full-Stokes models down to spatial scales as small as 10 km. Goldberg (2011) also showed that the DIVA was nearly as simple to implement as the hybrid SIA/SSA approximation, and Robinson et al. (2021) showed that it was significantly faster than other vertically integrated schemes at high (< 5 km) resolutions, due to the superior numerical stability and consequently larger time steps. Despite these advantages, the DIVA is, to our knowledge, currently only used in CISM (Lipscomb et al., 2019) and Yelmo (Robinson et al., 2021).

In this paper we present a detailed description of IMAU-ICE v2.0 and the benchmarking experiments performed. Sect. 2 provides a general description of IMAU-ICE, and describes the implementation of the DIVA in IMAU-ICE. In Sect. 3 we compare model results to several analytical solutions, demonstrating that the numerical solvers work as intended. In Sect. 4 we present our results of the EISMINT-I (Huybrechts et al., 1996), ISMIP-HOM (Pattyn et al., 2008), MISMIP (Pattyn et al., 2012), MISMIP+ (Asay-Davis et al., 2016; Cornford et al., 2020), and ABUMIP (Sun et al., 2020) intercomparison exercises. These experiments are listed briefly in Table 1. In Sect. 5 we conclude with a discussion of the relative merits of IMAU-ICE with respect to other widely-used ice-sheet models, and of the various improvements that we have planned for IMAU-ICE in the foreseeable future.

**Table 1: Benchmark experiments performed with IMAU-ICE. Each experiment aims to verify a different model component, at a different (range of) resolution(s), and over a different timespan.**

| Experiment | Model component | Resolution | Timespan |
|---|---|---|---|
| Halfar dome | SIA, ice thickness integration | 50 km | 100,000 yr |
| Bueler dome | SIA, ice thickness integration | 50 km | 10,000 yr |
| Schoof ice stream | SSA | 40 km – 10 km | n/a |
| EISMINT-I | SIA, ice thickness integration, thermodynamics | 50 km | 120,000 yr |
| ISMIP-HOM | SIA/SSA, DIVA | 2 km – 62.5 m | n/a |
| MISMIP | Grounding-line migration | 40 km – 10 km | 45,000 yr |
| MISMIP+ | Grounding-line migration, buttressing | 5 km – 2 km | 200 yr |

| ABUMIP | Total model | | 40 km – 10 km | 500 yr |
|--------|-------------|--|---------------|--------|

## 2 Model description

### 2.1 General model description

IMAU-ICE v2.0 by default solves the DIVA approximation to the stress balance to find the englacial velocity field; other stress-balance approximations (SIA, SSA, hybrid SIA/SSA) are available for specific experiments and testing. The mass
conservation equation is integrated through time using an explicit solver; a semi-implicit solver is available, offering improved numerical stability at an increased computational expense. The model has a dynamic time-step, which is calculated using a predictor/corrector method to provide an estimate of the truncation error in the ice thickness (Cheng et al., 2016). The implementation of this method is adopted from Yelmo (Robinson et al., 2020). The model is thermomechanically coupled; the vertical coordinate is discretised using an irregularly-spaced scaled coordinate, and the heat equation is solved on the resulting
3-D grid. The version of the heat equation that is solved by the model includes horizontal and vertical advection, vertical (but not horizontal) diffusion, strain heating from vertical shearing, and a spatially variable geothermal heat flux (by default obtained from Shapiro and Ritzwoller, 2004) at the base of grounded ice. A derivation of this equation and its discretisation is provided by Berends et al. (2021b). The englacial temperature is used to determine the flow factor for Glen's flow law using an Arrhenius relation, following Huybrechts (1992). Anisotropic rheology and ice damage are not accounted for in the model,
but Glen's flow law factor can be scaled separately for grounded and floating ice using flow enhancement factors; unless otherwise specified, these are set to unity in all experiments presented here.

IMAU-ICE v2.0 is suitable for both future projections of Greenland and Antarctica, as well as for simulations of glacial cycles. For the latter purpose, the model can simultaneously simulate the evolution of ice-sheets in four model regions: North America,
Eurasia, Greenland, and Antarctica, shown in Fig. 1. To prevent double-counting, ice thickness is kept zero in the Greenlandic parts of the North American and Eurasian model regions, or in the Icelandic parts of the Greenland region. The grid resolution is fully configurable by the user. For palaeoglaciological applications, we typically use 40 km everywhere except for Greenland, where 20 km is used (made feasible by that ice sheet's relatively small size).

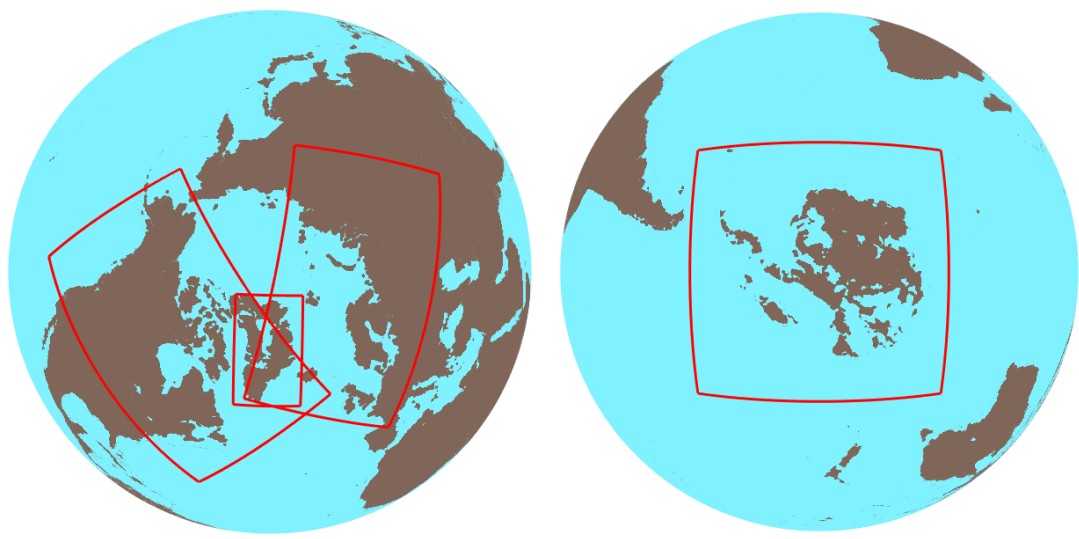

**Figure 1: The four model regions of IMAU-ICE. No ice is shown; brown (blue) indicates bedrock above (below) present-day sea level.**

## 2.2 Ice dynamics: the depth-integrated viscosity approximation

5     **Table 2: Model symbols, units, and default values where applicable**

| Symbol | Description | Units | Value |
|:---:|:---|:---|:---:|
| $A$ | Glen's flow law factor | Pa$^{-n}$ yr$^{-1}$ | |
| $b$ | Bedrock elevation | m | |
| $b_{min}$ | Lower bedrock elevation threshold for till friction angle | m | -1000 |
| $b_{max}$ | Upper bedrock elevation threshold for till friction angle | m | 0 |
| $\beta$ | Basal friction coefficient | Pa m$^{-1}$ yr | |
| $\beta_{\text{eff}}$ | "Effective" basal friction coefficient (including vertical shear) | Pa m$^{-1}$ yr | |
| $d_{sat}$ | Water depth where pore saturation occurs | m | 1000 |
| $d_w$ | Water depth | M | |
| $\dot{\varepsilon}_e$ | Effective strain rate | yr$^{-1}$ | |
| $\varphi$ | Till friction angle | degrees | |
| $\varphi_{min}$ | Minimum till friction angle (when $b = b_{min}$) | degrees | 5 |
| $\varphi_{max}$ | Maximum till friction angle (when $b = b_{max}$) | degrees | 20 |
| $g$ | Gravitational acceleration | m s$^{-2}$ | 9.81 |
| $H$ | Ice thickness | m | |
| $\eta$ | Effective viscosity | Pa yr | |

| | | | | |
|---|---|---|---|---|
| $n$ | Glen's flow law exponent | | | 3 |
| $p_w$ | Pore-water pressure | | Pa | |
| $q$ | Exponent in regularised Coulomb friction law | | | 0.3 |
| $\rho_i$ | Density of ice | | kg m$^{-3}$ | 910 |
| $\rho_w$ | Density of seawater | | kg m$^{-3}$ | 1028 |
| $s$ | Surface elevation | | m | |
| $\boldsymbol{\tau}_b$ | Basal shear stress vector | | Pa | |
| $\tau_{b,x}$ | Basal shear stress in x-direction | | Pa | |
| $\tau_{b,y}$ | Basal shear stress in y-direction | | Pa | |
| $\tau_c$ | Basal yield stress | | Pa | |
| $\boldsymbol{\tau}_d$ | Driving stress vector | | Pa | |
| $\tau_{d,x}$ | Driving shear stress in x-direction | | Pa | |
| $\tau_{d,y}$ | Driving shear stress in y-direction | | Pa | |
| $\boldsymbol{u}$ | Horizontal ice velocity vector | | m yr$^{-1}$ | |
| $u$ | Horizontal ice velocity in x-direction | | m yr$^{-1}$ | |
| $\boldsymbol{u}_b$ | Horizontal ice-basal velocity vector | | m yr$^{-1}$ | |
| $u_0$ | Threshold velocity in regularised Coulomb sliding law | | m yr$^{-1}$ | 100 |
| $v$ | Horizontal ice velocity in y-direction | | m yr$^{-1}$ | |
| $w$ | Vertical ice velocity | | m yr$^{-1}$ | |
| $z_{SL}$ | Sea-level elevation | | m | |

As given by Goldberg (2011), the equations for the DIVA stress balance read:

$$\frac{\partial}{\partial x}\left[2\bar{\eta}H\left(2\bar{u}_x + \bar{v}_y\right)\right] + \frac{\partial}{\partial y}\left[\bar{\eta}H\left(\bar{u}_y + \bar{v}_x\right)\right] - \beta_{\text{eff}}\bar{u} = -\tau_{d,x}. \tag{1a}$$

$$\frac{\partial}{\partial y}\left[2\bar{\eta}H\left(2\bar{v}_y + \bar{u}_x\right)\right] + \frac{\partial}{\partial x}\left[\bar{\eta}H\left(\bar{v}_x + \bar{u}_y\right)\right] - \beta_{\text{eff}}\bar{v} = -\tau_{d,y}. \tag{1b}$$

The symbols appearing in these and other equations are listed in Table 2. Subscript notation is used to indicate derivatives, e.g. $u_x = \frac{\partial u}{\partial x}$. Bars indicate vertical averages, e.g. $\bar{u} = \frac{1}{H}\int_b^s u(z)dz$. The two square-bracket terms on the left-hand side

5  represent the "membrane stresses" from longitudinal stretching and lateral shearing (also present in the SSA), while the term $\beta_{\text{eff}}\bar{u}$ represents both the frictional shear stress at the base (also present in the SSA), and the viscous stress from vertical shearing in the ice (also present in the SIA). These stresses are balanced by the gravitational driving stress on the right-hand

side. The vertically averaged effective viscosity $\bar{\eta}$ is determined as a function of the temperature-dependent flow factor $A$ and the effective strain rate $\dot{\varepsilon}_e$:

$$\bar{\eta} = \frac{1}{H}\int_b^s \eta(z)dz = \frac{1}{H}\int_b^s \frac{1}{2}A^{\frac{-1}{n}}\dot{\varepsilon}_e^{\frac{1-n}{n}}dz \tag{2}$$

$$\dot{\varepsilon}_e^2 = \bar{u}_x^2 + \bar{v}_y^2 + \bar{u}_x\bar{v}_y + \frac{1}{4}\left[\bar{u}_y + \bar{v}_x\right]^2 + \frac{1}{4}\bar{u}_z + \frac{1}{4}\bar{v}_z \tag{3}$$

The term $\beta_{\text{eff}}$ is related to the basal friction term $\beta$, but includes an additional term that accounts for the vertical shear stress:

$$\beta_{\text{eff}} = \frac{\beta}{1 + \beta F_2} \tag{4}$$

$$F_n = \int_b^s \frac{1}{\eta(z)}\left(\frac{s-z}{H}\right)^n dz \tag{5}$$

A comprehensive derivation of these equations is provided by Lipscomb et al. (2019). The way the DIVA is solved numerically
in IMAU-ICE v2.0 is mostly adopted from Yelmo (Robinson et al., 2021), and is very similar to typical SSA solvers, using an "outer loop" where the effective viscosity, the effective basal friction, and the velocities are iteratively updated until the solution converges. The stress balance discretisation on the staggered Arakawa-C grid is described in Appendix A.

Rather than applying stress boundary conditions at the ice margin, IMAU-ICE follows the "infinite slab" approach used by
e.g. Ritz et al. (2001) and Pattyn (2017), where ice-free pixels are still assigned a viscosity (as if there were a very thin layer of ice present), so that the velocity equations are solved over the entire model domain. At the domain boundary, a simple Neumann boundary condition is prescribed, so that velocities on boundary grid cells are equal to those on next-to-boundary grid cells.

## 2.3 Sliding, grounding-line migration, and calving

As in earlier model versions, IMAU-ICE v2.0 uses a regularised Coulomb-type sliding law similar to Pattyn (2017), where the till friction angle and basal yield stress are calculated following Martin et al. (2011). Here, the basal friction $\boldsymbol{\tau}_b$ is related to the basal velocity $\boldsymbol{u}_b$ and the basal yield stress $\boldsymbol{\tau}_c$:

$$\boldsymbol{\tau}_b = \tau_c \frac{|\boldsymbol{u}_b|^{q-1}\boldsymbol{u}_b}{u_0^q}. \tag{6}$$

This results in the following expression for the sliding term $\beta$:

$$\beta = \frac{\tau_c|\boldsymbol{u}_b|^{q-1}}{u_0^q}. \tag{7}$$

The basal yield stress $\tau_c$ is related to the pore water pressure $p_w$ and the till friction angle $\varphi$:

$$\tau_c = \tan\varphi\,(\rho_i gH - p_w) \tag{8}$$

Here, the term between brackets is the effective overburden pressure. The till friction angle and pore water pressure are parameterised respectively as functions of the bedrock elevation $b$ and the water depth $d_w = z_{SL} - b$:

$$\varphi = w_b \varphi_{max} + (1 - w_b)\varphi_{min}, \tag{9}$$

$$w_b = \min\left(1, \max\left(0, \frac{b - b_{min}}{b_{max} - b_{min}}\right)\right), \tag{10}$$

$$p_w = 0.96\rho_i g H \lambda_w, \tag{11}$$

$$\lambda_w = \min\left(1, \max\left(0, \frac{d_w}{d_{sat}}\right)\right). \tag{12}$$

The default values of the different constants (all of which can be changed at run-time in IMAU-ICE through the configuration file) are listed in Table 1. These parameters are uniform over the model domain; the spatial variability in the friction coefficient

is introduced by the bedrock elevation. The scaling coefficients $w_b$ and $\lambda_w$ are limited between 0 and 1. Eq. 11 describes a purely local relation between subglacial water pressure and ice thickness; including a more elaborate description of subglacial hydrology will be a part of future work.

In order to accurately reproduce grounding-line migration, IMAU-ICE follows the approach used in PISM (Feldmann et al., 2014) and in CISM (Leguy et al., 2021) by scaling the basal friction term $\beta_{eff}$ near the grounding line with the sub-grid

grounded fraction. The sub-grid grounded fraction is found by bilinearly interpolating the thickness above floatation, using the analytical solution derived by Leguy et al. (2021). As we will show in Sect. 4.3, this approach results in grounding-line hysteresis smaller than the grid resolution in the MISMIP experiment. Preliminary experiments with the grounding-line flux condition that was used in IMAU-ICE v1.0 (following the approach of Pollard and DeConto, 2012, and Pattyn, 2017) showed that this approach yields similar results in the schematic MISMIP experiments, but made it more difficult to maintain numerical

stability in realistic applications.

We note that our approach differs from that of PISM (Feldmann et al., 2014) and CISM (Lipscomb et al., 2019) in that we scale $\beta_{eff}$ with the square of the grounded fraction, rather than with the grounded fraction itself. We found that this yields less ice-sheet asymmetry and grounding-line hysteresis in the MISMIP experiment. The discrepancy might be caused by the fact that PISM uses one-sided differencing to calculate the driving stresses in next-to-grounding-line grid cells (they define SSA

velocities on the regular grid, in contrast to our staggered approach). In CISM, velocities are defined on the double-staggered Arakawa B-grid, possibly introducing some additional numerical diffusion when calculating the fluxes in the ice thickness integration.

Calving is parameterised in IMAU-ICE v2.0 by a simple threshold-thickness calving law, with a default threshold thickness

of 200 m.

## 2.4 Glacial isostatic adjustment

Two options for glacial isostatic adjustment (GIA) are included: a simple ELRA (Elastic Lithosphere, viscously Relaxed Asthenosphere) model with eustatic sea-level change, and the more elaborate sea-level equation solver SELEN (Spada and Stocchi, 2007), which includes the self-gravitating effects of both ice and ocean loading for all four ice sheets, coastline migration, and rotational feedback, allowing the calculation of relative sea level changes. SELEN solves these equations globally using spherical harmonics. The coupling to SELEN was first set up by de Boer et al. (2014) for ANICE, and has been restructured for IMAU-ICE v2.0 to provide more flexibility and user-friendliness. The code of SELEN has also been parallelised, so that including SELEN in a 120-kyr glacial cycle simulation (with a coupling interval of 1 kyr) now adds only about 1 wall clock hour (24 core hours) when running the simulation at a spectral resolution of harmonic degree 64 on a 24-core system. At harmonic degree 128, this increases to 6 wall clock hours (144 core hours).

## 2.5 Climate and mass balance

IMAU-ICE v2.0 by default uses the climate matrix method from Berends et al. (2018) to calculate the surface climate. In this approach, output from simulations of the pre-Industrial (PI) and Last Glacial Maximum (LGM) climate from the HadCM3 GCM (Singarayer and Valdes, 2010) is combined using spatially variable weights that depend on externally prescribed $CO_2$ and internally modelled ice-sheet geometry. This method reproduces the general features of the ice-albedo feedback, the elevation-temperature feedback, and the orographic forcing of precipitation, resulting in a modelled climate that is mutually consistent with the modelled ice-sheet geometry. Previous work by Berends et al. (2018) shows that this method resulted in simulated ice-sheet geometries at the LGM that agreed significantly better with geomorphological evidence than those from more simplistic climate index forcing methods. The option to use a prescribed climate or SMB forcing is included, which is useful for future projections or schematic experiments.

The surface mass balance (SMB) is obtained from the calculated/prescribed climate using the insolation-temperature model IMAU-ITM (Berends et al., 2018). This model uses parameterisations to partition precipitation into snow and rain, calculate snow melt as a function of insolation and surface temperature, calculate refreezing, and calculate the albedo. IMAU-ITM participated in the recent GrSMBMIP intercomparison exercise (Fettweis et al., 2020), where it was shown to perform well at simulating the recent mass balance of Greenland, at a very low computational cost.

The basal mass balance is by default calculated using the sub-shelf melt parameterisation by Martin et al. (2011), combined with the glacial/interglacial parameterisation and the subtended-angle/distance-to-open-ocean parameterisation by Pollard and DeConto (2009). Here, the sub-shelf melt rate $S$ is calculated using a linear relation to the thermal forcing:

$$S = \rho_w c_p \gamma_T \frac{T - T_f}{H_f \rho_i}. \tag{13}$$

The applied melt rate $M$ is calculated by interpolating between $S$ and the spatially uniform, temporally variable melt rates for exposed shelves $M_x$ and deep ocean $M_d$:

$$M = (1 - w_d)[(1 - w_x)S + w_x M_x] + w_d M_d. \tag{14}$$

The weighting factors for exposed shelves $w_x$ and deep ocean $w_d$ are calculated based on the water depth $d_w$, the widest subtended angle to the open ocean $\alpha_s$, and the shortest linear distance to the open ocean $r_o$:

$$w_x = \frac{\alpha_s - 80°}{30°} e^{-0.4 r_o}, \tag{15}$$

$$w_d = \frac{d_w - 1800\,m}{200\,m}. \tag{16}$$

The spatially uniform, temporally variable melt rates for exposed shelves $M_x$ and deep ocean $M_d$, as well as the temperature $T$ of the ocean water underneath the shelf in Eq. 13, are calculated from a set of reference values, using the glacial–interglacial variance parameterization by Pollard and DeConto (2009):

$$M_x = \begin{cases} w M_{x,\mathrm{PD}} + (1 - w)M_{x,\mathrm{cold}} & \text{if } w < 1, \\ (2 - w)M_{x,\mathrm{PD}} + (w - 1)M_{x,\mathrm{warm}} & \text{otherwise,} \end{cases} \tag{17}$$

$$M_d = \begin{cases} w M_{d,\mathrm{PD}} + (1 - w)M_{d,\mathrm{cold}} & \text{if } w < 1, \\ (2 - w)M_{d,\mathrm{PD}} + (w - 1)M_{d,\mathrm{warm}} & \text{otherwise,} \end{cases} \tag{18}$$

$$T = \begin{cases} w T_{\mathrm{PD}} + (1 - w)T_{\mathrm{cold}} & \text{if } w < 1, \\ (2 - w)T_{\mathrm{PD}} + (w - 1)T_{\mathrm{warm}} & \text{otherwise.} \end{cases} \tag{19}$$

The weighting factor $w$ is calculated based on the changes in the global annual mean surface temperature anomaly $T_s$ and the insolation at the top of the atmosphere $Q$:

$$w = 1 + \frac{T_s - T_{s,\mathrm{PD}}}{12\,K} + \frac{Q - Q_{\mathrm{PD}}}{40\,W\,m^{-2}} \tag{20}$$

The reference values for the melt rates and ocean temperature, which are listed in Table 3, were tuned by de Boer et al. (2013) to produce realistic present-day Antarctic shelves and grounding lines. Alternatively, the option to use a spatially uniform sub-shelf melt rate is included, which is useful for e.g. the ABUMIP experiments.

Table 3: Reference values for the uniform melt rates for exposed shelves $M_x$ and deep ocean $M_d$, and for the sub-shelf ocean temperature $T$.

| Parameter | Description | Units | Reference value |
|---|---|---|---|
| $M_{x,\mathrm{cold}}$ | Melt rate for exposed shelves in cold climate | m yr$^{-1}$ | 0 |
| $M_{x,\mathrm{PD}}$ | Melt rate for exposed shelves in present-day climate | m yr$^{-1}$ | 3 |
| $M_{x,\mathrm{warm}}$ | Melt rate for exposed shelves in warm climate | m yr$^{-1}$ | 6 |
| $M_{d,\mathrm{cold}}$ | Melt rate deep ocean in cold climate | m yr$^{-1}$ | 2 |
| $M_{d,\mathrm{PD}}$ | Melt rate deep ocean in present-day climate | m yr$^{-1}$ | 5 |
| $M_{d,\mathrm{warm}}$ | Melt rate deep ocean in warm climate | m yr$^{-1}$ | 10 |
| $T_{\mathrm{cold}}$ | Sub-shelf ocean temperature in cold climate | °C | -5 |

| $T_{PD}$ | Sub-shelf ocean temperature in present-day climate | °C | -1.7 |
| $T_{warm}$ | Sub-shelf ocean temperature in warm climate | °C | 2 |

Sub-shelf melt is by default applied only to grid cells floating at the centre, using the "floatation criterion melt parameterisation" (FCMP) scheme formulated by Leguy et al. (2021). The "no melt parameterisation" (NMP) and "partial melt parameterisation" (PMP) schemes are included as options; preliminary experiments in the MISMIP+ and ABUMIP 5 configurations showed that these schemes result in significantly more dependence on the grid resolution than the FCMP scheme.

## 3 Analytical solutions

In this section we present results from a number of schematic experiments that have analytical solutions. These concern only the SIA and SSA; no analytical solution to the DIVA has yet been derived. However, since the numerical solvers for the SSA 10 and the DIVA are nearly identical, proving that the SSA is solved correctly provides confidence that our DIVA solver is also functioning properly. This will be confirmed in Sect. 4, where we perform the ISMIP-HOM benchmark experiments.

### 3.1 Shallow ice approximation

Halfar (1981) derived an analytical solution to the SIA for the case of a radially symmetrical, isothermal ice sheet on a flat bed with zero mass balance. Since the ice sheet evolves only through ice dynamics, this is a useful experiment for verifying ice-15 sheet model numerics. Bueler et al. (2005) extended this solution to include a parameterised mass balance term. Comprehensive descriptions of these experiment and their analytical solutions are provided by Berends et al. (2021b). The ice-margin errors as function of model resolution for both experiments as simulated by IMAU-ICE v2.0 are shown in Fig. 2, with log-linear curves fitted to both sets of results. Both experiments show a convergence of ice-margin position error with model resolution of approximately the first order, indicating that the numerical schemes used to solve the SIA and integrate the ice thickness 20 equation are valid.

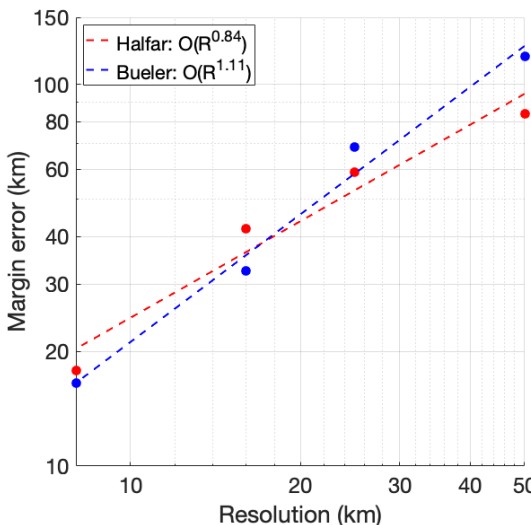

**Figure 2: The error in the simulated ice-margin position as a function of horizontal model resolution for the Halfar and Bueler dome experiments.**

## 3.2 Shallow shelf approximation

Schoof (2006) derived an analytical solution to the SSA for the case of an ice slab on a sloping bed, with a narrow band of lower friction running down the slope, resulting in the formation of an ice stream. This experiment was used to verify the numerical solver of PISM by Bueler and Brown (2009). Here we have adapted the experimental set-up to result in a wider ice-stream, so that the low-friction channel is still resolved when using a 40 km resolution. The parameters describing our set-up are listed in Table 4. We performed this experiment at resolutions ranging from 40 km to 10 km. The results are shown in Fig.

3; as can be seen, the simulated velocities converge towards the analytical solution with increasing resolution with order 1.7, close to the value of 1.9 reported by Bueler and Brown (2009) for their SSA solver. The error in the 40 km simulation reaches nearly 350 m/yr, or about 15 % of the analytical solution. This underlines the consensus that, while such a resolution is sufficient to accurately simulate large-scale ice-sheet dynamics, a higher resolution is desirable when the aim is to study smaller features such as single outlet glaciers.

**Table 4: Parameters for the SSA ice-stream experiment.**

| Parameter | Description | Value | Units |
|---|---|---|---|
| A | Glen's flow law factor | $10^{-18}$ | $Pa^{-3} \, yr^{-1}$ |
| n | Glen's flow law exponent | 3 | |
| $\frac{\partial b}{\partial x}$ | Bedrock slope | $3 \cdot 10^{-3}$ | |
| H | Ice thickness | 2,000 | m |
| L | Ice-stream half-width | 150 | km |

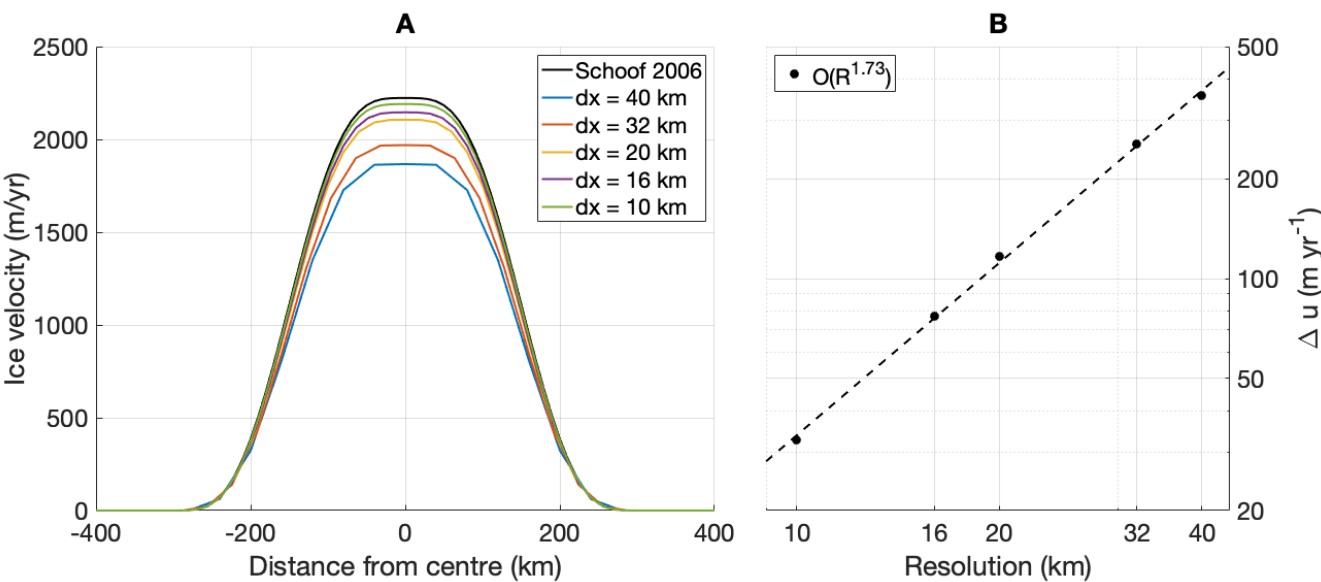

Figure 3, panel A: Cross-slope transect of the downslope velocity in the Schoof (2006) ice-stream experiment at different resolutions. Panel B: error in the mid-stream velocity versus grid resolution, showing a convergence of order 1.73.

## 4 Model intercomparison

In this section we perform experiments from the EISMINT-I (Huybrechts et al., 1996), ISMIP-HOM (Pattyn et al., 2008), MISMIP (Pattyn et al., 2012), MISMIP+ (Asay-Davis et al., 2016; Cornford et al., 2020), and ABUMIP (Sun et al., 2020) model intercomparison projects, to demonstrate that IMAU-ICE performs adequately when compared to other ice-sheet models. The DIVA is used in all of these experiments, except for EISMINT-I, which considers only the SIA.

### 4.1 EISMINT-I

The first EISMINT intercomparison exercise (Huybrechts et al., 1996) consists of six schematic experiments (see Table 5) similar to the Halfar dome, but with a parameterised mass balance that is independent of ice geometry. In experiments a and d, the mass balance is positive in the centre of the domain and decreases away from the ice divide, yielding a circular steady-state ice sheet. In experiments b and e, a 20 kyr sinusoid term is added to the mass balance to represent glacial cycles; in experiments c and f, the period is increased to 40 kyr. Experiments a, b, and c have a moving margin, achieved by prescribing the mass balance such that even at glacial maxima, the ice margin does not reach the edge of the domain. In experiments d, e, and f, the mass balance is increased such that even at glacial minima, the ice margin should lie outside the domain; a zero ice thickness boundary condition is prescribed at the domain boundary instead, leading to a fixed margin. These experiments

include uncoupled thermodynamics; the englacial temperature is calculated, but does not affect the ice flow factor (which is spatially and temporally constant). All of these experiments were performed at the original EISMINT-I resolution of 50 km. Fig. 4 shows the thickness at the ice divide for the four "glacial cycle" experiments (b, c, e, and f). Fig. 5 shows the simulated ice temperature at the base of the ice divide relative to the pressure melting point for the same set of experiments. For all four experiments, we find glacial-interglacial differences for both ice thickness and basal temperature that lie within, or very slightly outside of, the range of values reported by Huybrechts et al. (1996).

**Table 5: The six different EISMINT-I experiments.**

| Experiment | Margin | Mass Balance |
|---|---|---|
| a | moving | steady-state |
| b | moving | 20 kyr |
| c | moving | 40 kyr |
| d | fixed | steady-state |
| e | fixed | 20 kyr |
| f | fixed | 40 kyr |

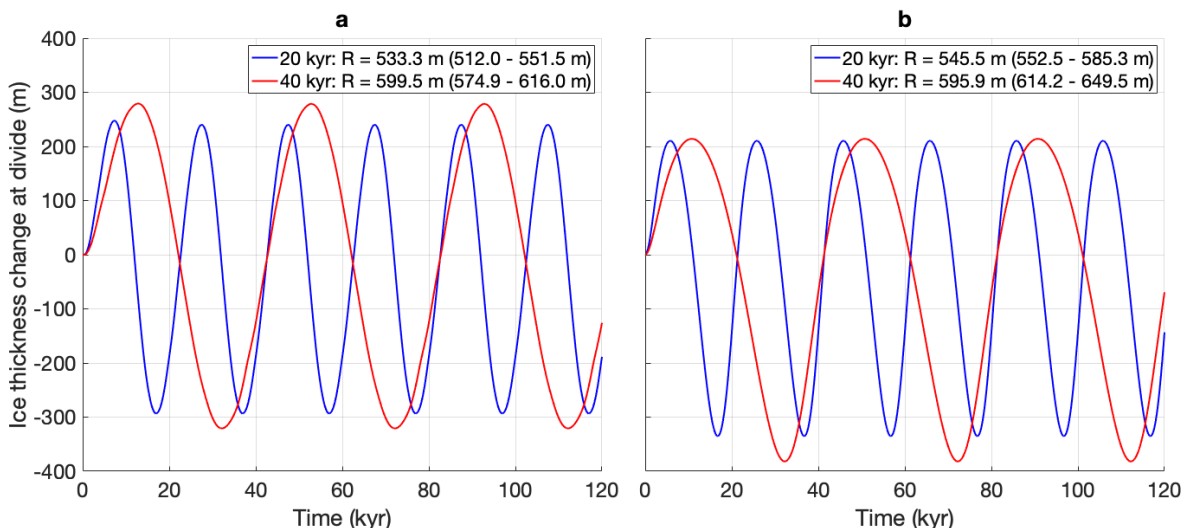

**Figure 4: Ice thickness at the divide over time for the "glacial cycle" experiments from the first EISMINT intercomparison exercise, with a moving margin (panel a) and a fixed margin (panel b). The legends list the simulated glacial-interglacial difference R for the last cycle, with the range of numbers reported by Huybrechts et al. (1996) listed between brackets for comparison.**

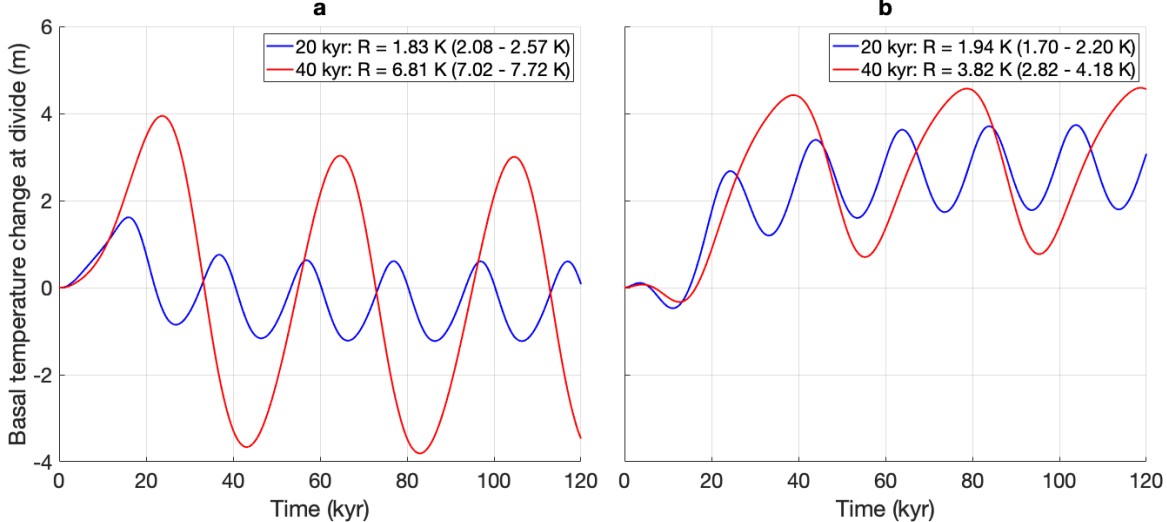

**Figure 5: Ice temperature at the base of the ice divide, relative to the pressure melting point, over time for the "glacial cycle" experiments from the first EISMINT intercomparison exercise, with a moving margin (panel a) and a fixed margin (panel b). The legends list the simulated glacial-interglacial difference R for the last cycle, with the range of numbers reported by Huybrechts et al. (1996) listed between brackets for comparison.**

## 4.2 ISMIP-HOM

We verify our DIVA solver by performing the six experiments from the ISMIP-HOM intercomparison exercise (Pattyn et al., 2008), which are listed in Table 6. Experiments A-D consist of calculating instantaneous ice velocities for a given schematic geometry, experiment E entails calculating the velocity profile along a flowline of an actual glacier, and experiment F consists of determining the steady-state geometry of an idealised ice sheet. Experiments A-D describe an ice slab on a sloping bed, which is perturbed by small periodic perturbations to either the bed elevation or the bed friction. For each experiment, the horizontal scale of these perturbations is varied, ranging from 160 km to 5 km, while the ice thickness is kept constant at 1 km. The grid resolution is varied so that the grid always measures 81 by 81 grid cells; for the 160 km experiment, the resolution is 2 km, whereas for the 5 km experiment it is 62.5 m. A complete description of the experiments is given by Pattyn et al. (2008).

**Table 6: The six different ISMIP-HOM experiments.**

| Experiment | Description |
| --- | --- |
| A | Ice slab on a sloping bed with sinusoid bumps in both directions |
| B | Ice slab on a sloping bed with sinusoid bumps in one direction |
| C | Ice slab on a sloping bed with changing friction in both directions |
| D | Ice slab on a sloping bed with changing friction in one direction |
| E | Haut Glacier d'Arolla |
| F | Ice slab on a sloping bed with a single Gaussian bump |

We performed all six experiments with IMAU-ICE v2.0, at all spatial scales used in the original ISMIP-HOM publication, with both the SIA/SSA and the DIVA ice dynamics. The results of experiment A are shown in Fig. 6. The SIA/SSA results become increasingly inaccurate as the spatial scale of the experiment decreases, with the velocities differing from the full-Stokes solution by up to a factor ten. The results from the DIVA remain much closer to those of the higher-order and full-Stokes models, in agreement with the findings reported by Goldberg (2011). Similar Figures for the other five experiments are provided in Appendix B; the results for these experiments are qualitatively similar.

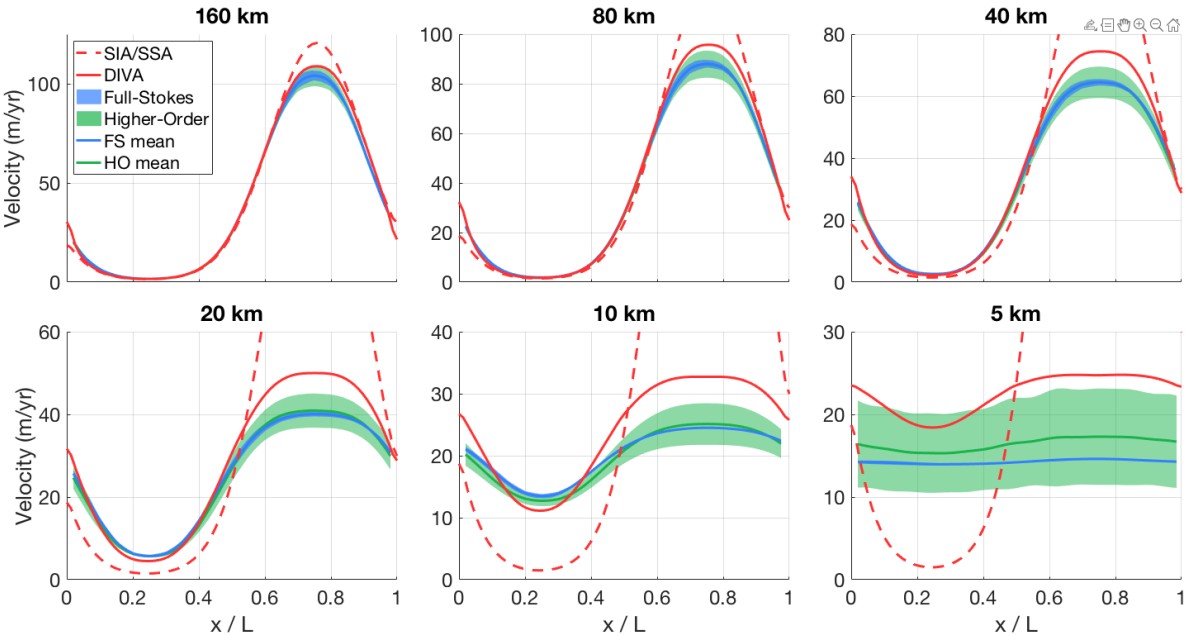

**Figure 6: Modelled surface velocity transects for all versions of ISMIP-HOM experiment A, calculated with both the old hybrid SIA/SSA solver (red dashed line) and the new DIVA solver (red solid line). The results of the higher-order models (green) and the full-Stokes models (blue) that participated in ISMIP-HOM are shown for comparison.**

## 4.3 Plan-view MISMIP

The first MISMIP experiment (Pattyn et al., 2012) was extended by Pattyn (2017) from a 1-D flowline to a 2-D plan-view setting, describing a cone-shaped island with a uniform positive mass balance. This results in a circular ice sheet, surrounded by an infinite ice shelf. In the experiment, after 15-kyr initialisation, the spatially uniform ice flow factor is subjected to 15-kyr step-wise decreases (increases), which result in an advancing (retreating) grounding line. We performed a single stepwise increase and decrease, with values of $10^{-16}$ and $10^{-17}$ for Glen's flow law factor, respectively. The results of this experiment, as performed with IMAU-ICE at resolutions of 40, 32, 20, 16, and 10 km, are shown in Fig. 7. As can be seen, grounding-line hysteresis decreases for higher resolution, and is smaller than the grid resolution for all five cases. The hysteresis in the 16 km simulation is, counter-intuitively, slightly larger than in the 20 km, but this difference is not significant as it is well below the model resolution.

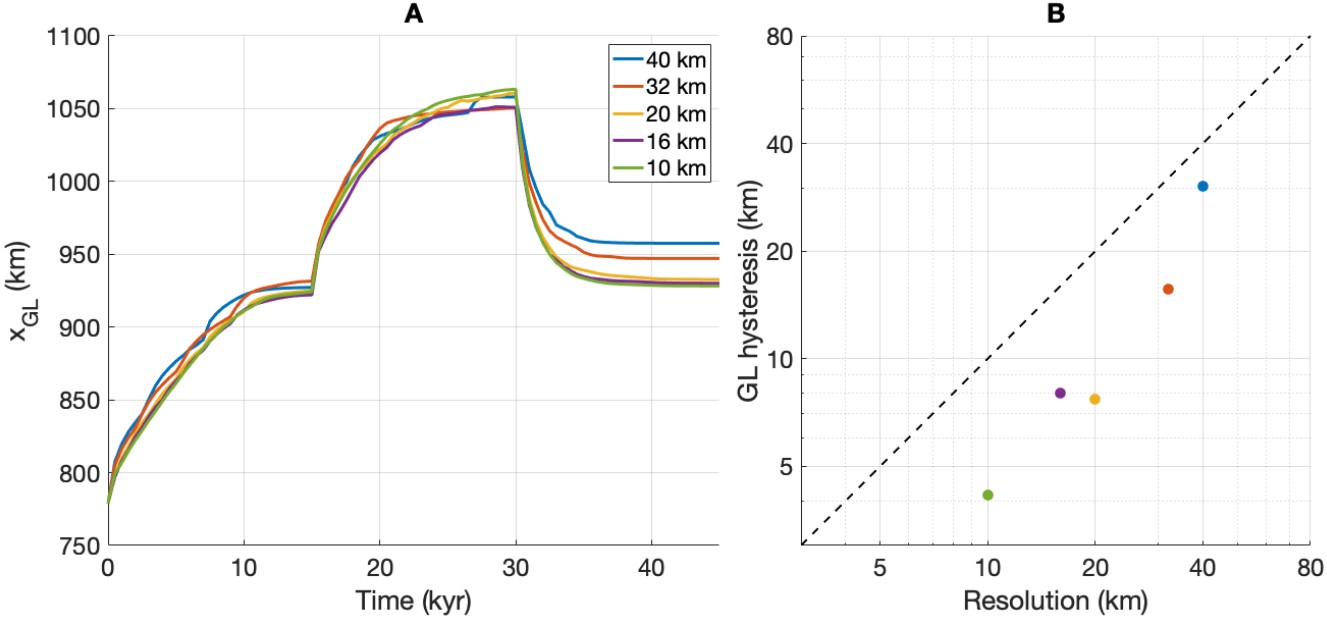

**Figure 7, panel A: Grounding-line position over time in the MISMIP experiment at different resolutions. The first 25 kyr show the initialisation phase. Panel B: grounding-line hysteresis (defined as the difference in grounding-line positions between t = 15 kyr and t = 45 kyr) versus model resolution.**

The 10 – 40 km resolutions we investigated here are much coarser than the values used by Feldmann et al. (2014) and Leguy et al. (2014), and we find proportionally stronger grounding-line hysteresis. However, since we still find values that are smaller than the grid resolution, we deem these errors to be acceptably small in the context of palaeo-ice-sheet modelling.

## 4.4 MISMIP+

The MISMIP+ experiment proposed by Asay-Davis et al. (2016) describes a laterally symmetrical ice sheet, flowing down an 800 km long, 80 km wide glacial valley into a confined bay. The geometry of the valley is such that the grounding line rests on a retrograde slope, kept in place by the buttressing forces of the confined shelf. The experimental protocol dictates that the mid-stream grounding-line position in steady state (achieved after a 20,000-yr spin-up) lies at exactly x = 450 km, which can be achieved by tuning Glen's flow law factor. After a steady state is achieved, several different forward experiments are carried out. For the first 100 years, a finite basal melt (which was set to zero during the spin-up) is prescribed in experiment ice1r, forcing the ice sheet to retreat. At t = 100 yr the experiment splits into two branches; experiment ice1rr continues the retreat for another 100 years, while experiment ice1ra stops the basal melt, allowing the ice sheet to advance again. Experiments ice2r, ice2rr, and ice2ra are similar, except that now a very high melt rate is applied near the ice front, mimicking an increase in calving. Experiment ice0 is a control run with no melt or calving throughout, such that the grounding line should not move. A complete description of the experimental set-up is provided by Asay-Davis et al. (2016); the results of the intercomparison were presented by Cornford et al. (2020).

We performed the MISMIP+ experiments with IMAU-ICE, at a resolution of 2 km, using a Weertman sliding law (with the uniform bed roughness value specified by Asay-Davis et al. (2016)), the DIVA solver, and the FCMP sub-grid melt scheme. The results of our simulations are compared to the model ensemble by Cornford et al. (2020) in Fig. 8. IMAU-ICE produces grounding-line positions that agree well with the ensemble, generally showing less retreat than the ensemble mean, but still

well within the ensemble range. Preliminary experiments showed that the rate of grounding-line retreat is sensitive to the combination of grid resolution and sub-grid melt scheme, with the FCMP scheme showing the least dependence on resolution. These results will be investigated in detail in future work.

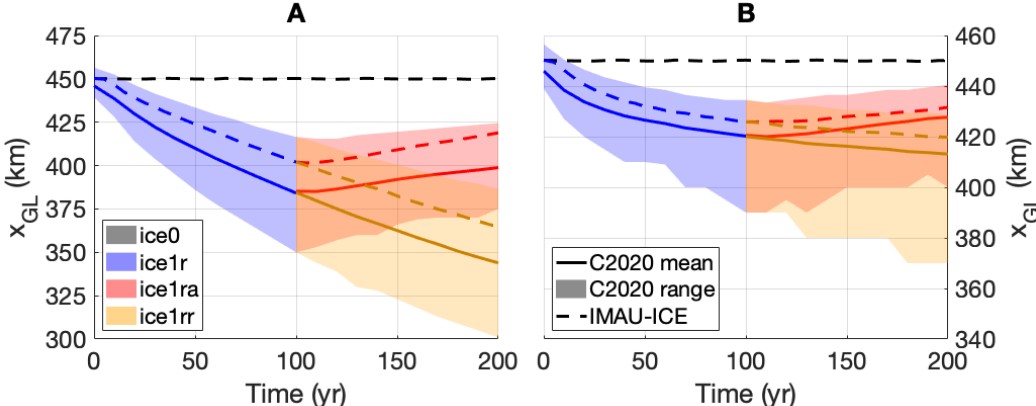

**Figure 8: Grounding-line position over time in the different MISMIP+ experiments. Panel A: experiments ice0 (black), ice1r (blue),**
**ice1ra (red), and ice1rr (gold). Panel B: experiments ice0 (black), ice2r (blue), ice2ra (red), and ice2rr (gold). The range and mean of the model ensemble by Cornford et al. (2020) are respectively shown by shaded areas and solid lines (labelled C2020 range and C2020 mean); the results of the IMAU-ICE simulations are shown by dashed lines. Note the different y-axis scales of the two panels.**

### 4.5 ABUMIP

The Antarctic Buttressing Model Intercomparison Project (ABUMIP; Sun et al., 2020) investigates the dynamic response of
the Antarctic ice sheet to the sudden disintegration of ice shelves, either by strongly increasing the sub-shelf melt (ABUM), or by forcibly removing all floating ice in the model (ABUK). Model drift is quantified in the control experiment (ABUC), where no forcing is applied; the experimental protocol does not require the participating models to be in a steady state at the start of the experiments. In all of these experiments, we chose to keep the SMB fixed to the present-day values simulated by the regional climate model RACMO2.3 (van Wessem et al., 2014), and mapped to a square grid using OBLIMAP 2.0 (Reerink et
al., 2010, 2016). We initialised the model with the observed present-day geometry from the Bedmachine Antarctica v1.0 dataset (Morlighem et al., 2019) and englacial temperatures according to the Robin solution. The spatially variable, temporally constant geothermal heat flux is prescribed based on the data from Shapiro and Ritzwoller (2004). We performed a simple spin-up consisting of three phases: (1) a short 100-yr unforced relaxation; (2) a 240,000-yr thermal spin-up with a fixed geometry; and (3) a 100,000-yr relaxation with fixed temperatures and fixed shelf geometry. The first phase serves to smooth
the geometry and reduce the flow gradients, improving the numerical stability of the heat equation in the second phase. The surface temperature forcing during the second phase (thermal spin-up) is derived from snapshots of the pre-industrial and of

the last glacial maximum produced with HadCM3 (Singarayer and Valdes, 2010), using a simple glacial-index method based on an ice-core $CO_2$ record (Bereiter et al., 2015) to interpolate between them. This ensures that the englacial temperatures of the spun-up ice sheet retains a cold glacial history. Lastly, during the third phase, the ice temperature and the thickness of floating ice shelves are kept constant, allowing the grounded ice to reach a steady state. We tuned the bed roughness parameters

5  and the flow enhancement factors to minimise the drift in ice volume during this phase. In order to achieve a no-drift steady state in the ABUC control experiment, appropriate basal melt needs to be prescribed. Since the basal melt formulation is irrelevant for the ABUM and ABUK forced experiments, we used a simple geometry-based inversion method to derive melt rates in ABUC, similar to the work of Bernales et al. (2017). Here, the melt rates are continuously updated throughout the simulation, increasing (decreasing) when the modelled shelf is too thick (thin). The results of all three experiments, simulated

10  at resolutions of 40, 32, 20, 16, and 10 km, are shown in Fig. 9, together with the results from all the models from Sun et al. (2020), including IMAU-ICE v1.0.

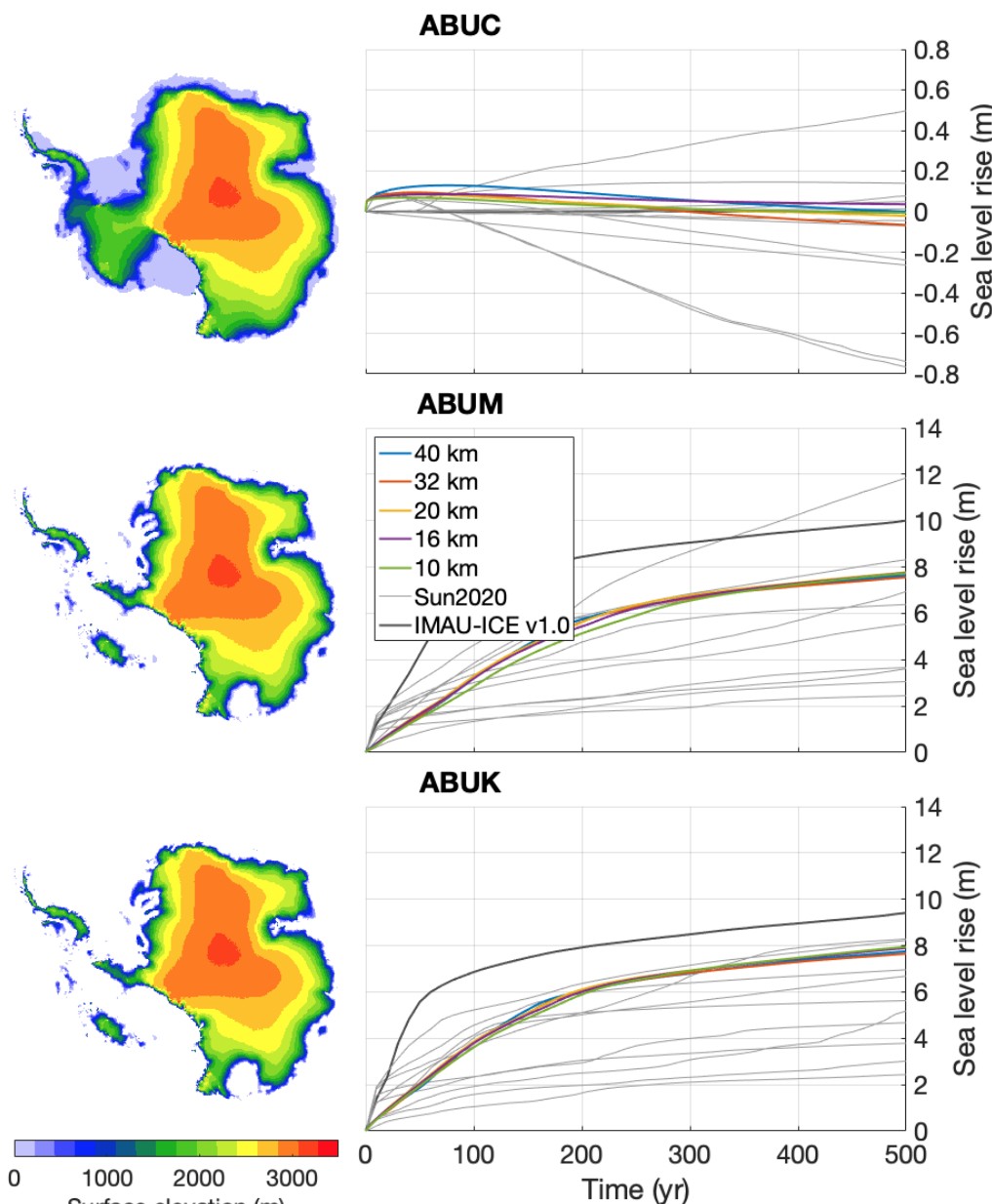

**Figure 9: Surface elevation after 500 model years (left) and sea-level rise over time (right) at different model resolutions in the ABUMIP experiments. Grey lines indicate the results of the models that participated in the ABUMIP model comparison (Sun et al., 2020); the results from IMAU-ICE v1.0 (run at 32 km, taken from Sun et al., 2020) are shown by the thick grey line. The panels on the left show results from the 10 km experiments for IMAU-ICE v2.0 at t = 500 yr.**

The ABUC control experiment, shown in top row of Fig. 9, displays a very small amount of drift of -0.1 to 0.1 m of sea-level rise after 500 years. The ABUM and ABUK experiments yield very similar sea-level curves, that do not significantly depend on model resolution in either experiment. The modelled sea-level rise at t = 500 yr is 7.6 – 7.9 m for both experiments, near

the upper end of the ensemble range of Sun et al. (2020). Although several models from the ensemble by Sun et al. (2020) show substantial differences between the two experiments, we expect to see little difference. Whether the floating ice is removed by calving (ABUK) or by melting (ABUM), the result should be the same (i.e. no shelves). Although Sun et al. (2020) mention that in ABUM a very small shelf could remain, even in the extreme case of a grounding-line thickness and velocity

of 2,000 m and 1,000 m/yr, respectively, this would result in a shelf of only 2.5 km long, which we do not expect would lead to any significant buttressing.

Compared to IMAU-ICE v1.0, v2.0 produces a slower retreat during the first 100 years of the experiment, which is mostly due to contributions from West Antarctica. During the following 400 years, where most of the contribution is from East Antarctica,

retreat rates are generally the same in both model versions. There are several differences between the two model versions that can explain these discrepancies, most importantly the approximation to the stress balance (hybrid SIA/SSA vs. DIVA), the treatment of the grounding line (flux condition vs. sub-grid friction scaling), and the spin-up procedure (10-kyr steady state in version 1.0 vs. 100-yr relaxation + 240-kyr thermal spin-up + 100-kyr relaxation in version 2.0). A more in-depth investigation of the effects of these choices on the modelled sea-level rise is ongoing, but is beyond the scope of this model description

paper.

## 5 Conclusions and discussion

### 5.1 Current status and applicability

We have presented version 2.0 of the vertically integrated ice-sheet model IMAU-ICE, which solves the DIVA approximation to the stress balance. We verified the numerical schemes used to solve the stress balance and integrate the ice thickness

equation. These yield results that match several analytical solutions, or results from other ice-sheet models used in intercomparison experiments. Our findings match those of Goldberg (2011), showing that the DIVA remains physically accurate at much smaller resolutions than the hybrid SIA/SSA. We have also replaced the grounding-line flux condition used in IMAU-ICE v1.0 with a sub-grid scaling of the basal friction near the grounding line, resulting in improved numerical stability, while still achieving good results in terms of grounding-line hysteresis and resolution dependence in the MISMIP

experiment. Overall, the benchmark experiments performed in this study indicate that the 40 km resolution typically used for palaeoglaciological applications produces reliable results in terms of large-scale ice-sheet evolution, with no appreciable model error when compared to higher-resolution (10 km) simulations. Errors caused by unresolved topographical features are a different matter, which is not studied in this project. Of course, when the aim is to study smaller-scale ice-sheet features on shorter time scales, a higher resolution is to be recommended; however, for the large-scale ice-sheet dynamics that are typically

of interest to palaeo-ice-sheet modellers, the small resolution-dependent errors reported here are much smaller than the errors in the forcing such as the palaeoclimate and the palaeogeography.

IMAU-ICE v2.0 can be used both for future projections and for palaeo-applications, but generally provides more support for the latter. With minimum effort, the user can change the external configuration file to choose between different (palaeo-)climatic and topographic conditions, geological periods, ice, methods of forcing, ice-dynamical approximations, and mass balance parameterisations. This ensures easy reproduction of results, as well as a smooth workflow. The climate component

of the model is particularly flexible; although previous palaeoglaciogical studies using ANICE/IMAU-ICE all used the same HadCM3 output (Singarayer and Valdes, 2010) to construct the climate matrix, the matrix method can in principle be applied to output from any GCM, and IMAU-ICE v2.0 has been designed to easily accommodate different GCM data. The climate data needs to be provided on a regular, global lon/lat-grid, and the projection to the ice-model grid is automatically performed internally. Taking advantage of this ease-of-use, IMAU-ICE is currently being used in different palaeoglaciological studies.

One example concerns the evolution of the Antarctic ice sheet during the warm Miocene (Stap et al., *in review*), using climate data from the GENESIS GCM, and an Antarctic palaeotopographic reconstruction. Another currently ongoing project involves an ensemble of simulations of the last glacial cycle, forced with all of the GCMs that participated in PMIP3 (Scherrenberg et al., *in prep.*).

Since about 2018, IMAU-ICE has also been used for simulations of near-future ice-sheet evolution. New features introduced in the code overhaul from ANICE to IMAU-ICE, including improved high-resolution support, prescribed climate/SMB-forcing, improved grounding-line dynamics, and easy restarting (to facilitate different spin-up strategies) have greatly improved the model's usability and applicability in such settings. However, compared to ice-sheet models that have been developed specifically for this purpose, IMAU-ICE still has relatively simplistic representations of physical processes such as

glacial rheology and damage, subglacial hydrology and basal sliding, and englacial stresses in areas with high aspect ratios. While this makes it feasible to perform large ensemble simulations at relatively coarse resolutions, it means the users must take more caution when interpreting model results on sub-basin spatial scales.

IMAU-ICE v2.0 is partly parallelised; the matrix equations representing the DIVA (the most computationally expensive part

of the model by far) are solved using the PETSc library (Balay et al., 2021), whereas all other routines are parallelised using MPI shared memory. This is a compromise between performance and user-friendliness; while the code structure and syntax of MPI shared memory are very similar to non-parallelised code, it is not easy to extend this to a fully distributed implementation. Conversely, PETSc is highly scalable, but since it is less friendly to novice ice-sheet modellers, its use has been limited to the velocity solver. As a result of this compromise, IMAU-ICE can be run only on the maximum number of processors on the

user's hardware system that can access the same physical memory chip (usually 16, 24, or 32 cores on typical scientific computation systems). For the long (> 100,000 yr), low-resolution (~40 km) palaeo-ice-sheet simulations, and short (< 1,000 yr), medium-resolution (~16 km) future projections that we intend to use the model for, this results in computation times that are typically short enough to run the model overnight.

## 5.2 Future research

In their recent study, Leguy et al. (2021) demonstrated the strong effect of sub-grid schemes of sub-shelf melt on grounding-line dynamics. They showed that the relative "performance" of these schemes (indicated by the dependence on grid resolution) varied between different choices of sliding law, melt parameterisation, and experimental set-up. Seroussi and Morlighem (2018) performed very similar experiments, yet found different results, underlining the uncertainty that still surrounds the treatment of sub-shelf melt near the grounding line. We are currently working on an in-depth investigation of these processes in IMAU-ICE. This includes both the schematic MISMIP+ geometry (Asay-Davis et al., 2016) used by Leguy et al. (2021) and the realistic Antarctic geometry, where in both settings we study the interplay between the choice of sliding law, sub-grid melt scheme, grid resolution, and stress balance approximation.

The parameterisations of sub-shelf melt and calving currently used in IMAU-ICE are overly simplistic. We are currently working on a thorough overhaul of these model components, which will include an implementation of the PICO model (Reese et al., 2018b), as well as a more elaborate plume model (Lazeroms et al., 2018). Before IMAU-ICE can be used to participate in intercomparison projects for future projections such as ISMIP, a better spin-up strategy is required. In order to make this possible, a basal inversion routine is currently being implemented in IMAU-ICE (Berends et al., *in prep*). Other ongoing work includes a more elaborate study of the feedbacks between ice-sheet geometry and climate using the matrix method, in the context of the Miocene (Stap et al., 2022) and the last glacial cycle (Scherrenberg et al., *in prep*).

## Code and data availability

The source code of IMAU-ICE is maintained on Github at https://github.com/IMAU-paleo/IMAU-ICE. The exact version used in this study (including makefiles, compiling scripts, run scripts, config files for all the simulations presented here, and Matlab scripts for creating the figures) is archived on Zenodo.org (doi: 10.5281/zenodo.5796152).

*Author contributions*. CJB wrote the code for the new model version, with contributions from LBS and HG. CJB performed the experiments and analysed the data. CJB wrote the draft of the manuscript; all authors contributed to the final version.

*Acknowledgements*. We are very grateful to Alex Robinson for his help in setting up the DIVA solver, and to Lars Zipf for his insightful comments on grounding-line dynamics.

*Competing interests*. The authors declare that they have no competing interests.

*Financial support*. This publication was supported by PROTECT. This project has received funding from the European Union's Horizon 2020 research and innovation programme under grant agreement no. 869304 (PROTECT; [article number

will be assigned upon acceptance for publication!]). The use of supercomputer facilities was sponsored by NWO Exact and Natural Sciences. Model runs were performed on the Dutch National Supercomputer Cartesius. we would like to acknowledge SurfSARA Computing and Networking Services for their support. L.B. Stap is funded by the Dutch Research Council (NWO), through VENI grant VI.Veni.202.031. Heiko Goelzer has received funding from the programme of the Netherlands Earth System Science Centre (NESSC), financially supported by the Dutch Ministry of Education, Culture and Science (OCW), under grant no. 024.002.001 and from the Research Council of Norway under projects INES (270061) and KeyClim (295046). High-performance computing and storage resources were provided by the Norwegian infrastructure for computational science through projects NN9560K, NN9252K, NS9560K, NS9252K and NS5011K.

## Appendix A – Discretisation

The ice-dynamical equations in IMAU-ICE are discretised using staggered Arakawa grids (see Fig. A1), a common practice in ice-sheet modelling. Material properties such as ice thickness, flow factor, and englacial temperature are defined on the regular Arakawa-A grid, while the horizontal velocity components $u,v$ are defined on the staggered Arakawa-Cx/Cy grids.

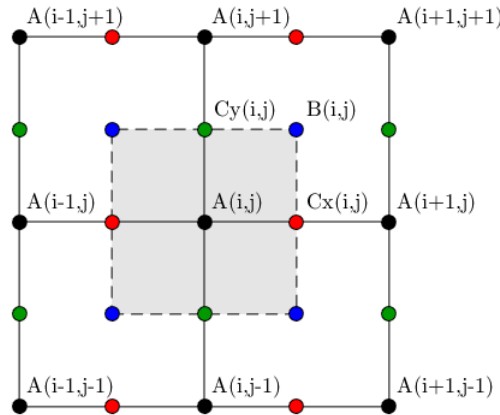

**Figure A1: The four Arakawa grids: regular (A; black), staggered in the x-direction (Cx; red), staggered in the y-direction (Cy; green), and staggered in both directions (B; blue).**

Following the approach from Yelmo (Robinson et al., 2020), Eqs. 1 are discretised by defining all derivatives as finite differences between the nearest half-grid points. Since the velocity $u$ is defined on the Cx-grid, the first outer derivative $\frac{\partial}{\partial x}[...]$ is defined as the difference between the neighbouring A-grid points, while the second outer derivative $\frac{\partial}{\partial y}[...]$ is defined as the difference between the neighbouring B-grid points:

$$\frac{\left[2\bar{\eta}H(2u_x + v_y)\right]_A^{i+1,j} - \left[2\bar{\eta}H(2u_x + v_y)\right]_A^{i,j}}{\Delta x} + \frac{\left[\bar{\eta}H(u_y + v_x)\right]_B^{i,j} - \left[\bar{\eta}H(u_y + v_x)\right]_B^{i,j-1}}{\Delta y} - \beta_{eff,Cx}^{i,j} u_{Cx}^{i,j} \quad \text{(A2)}$$

$$= -\tau_{d,x,Cx}^{i,j}.$$

The inner derivatives $u_x, u_y, v_x, v_y$ too, are discretised with respect to the nearest half-grid points, which are the Cx and Cy-grids where the velocities are defined:

$$u_{x,A}^{i+1,j} = \frac{u_{Cx}^{i+1,j} - u_{Cx}^{i,j}}{\Delta x}, u_{x,A}^{i,j} = \frac{u_{Cx}^{i,j} - u_{Cx}^{i-1,j}}{\Delta x}, u_{y,B}^{i,j} = \frac{u_{Cx}^{i,j+1} - u_{Cx}^{i,j}}{\Delta y}, u_{y,B}^{i,j-1} = \frac{u_{Cx}^{i,j} - u_{Cx}^{i,j-1}}{\Delta y},$$

$$v_{y,A}^{i+1,j} = \frac{v_{Cy}^{i+1,j} - v_{Cy}^{i+1,j-1}}{\Delta y}, v_{y,A}^{i,j} = \frac{v_{Cy}^{i,j} - v_{Cy}^{i,j-1}}{\Delta y}, v_{x,B}^{i,j} = \frac{v_{Cy}^{i+1,j} - v_{Cy}^{i,j}}{\Delta x}, v_{x,B}^{i,j-1} = \frac{v_{Cy}^{i+1,j-1} - v_{Cy}^{i,j-1}}{\Delta x}. \quad \text{(A3)}$$

Substituting Eqs. A3 into Eq. A2 yields:

$$\frac{2(\bar{\eta}H)_A^{i+1,j}}{\Delta x}\left[2\frac{u_{Cx}^{i+1,j} - u_{Cx}^{i,j}}{\Delta x} + \frac{v_{Cy}^{i+1,j} - v_{Cy}^{i+1,j-1}}{\Delta y}\right]$$

$$- \frac{2(\bar{\eta}H)_A^{i,j}}{\Delta x}\left[2\frac{u_{Cx}^{i,j} - u_{Cx}^{i-1,j}}{\Delta x} + \frac{v_{Cy}^{i,j} - v_{Cy}^{i,j-1}}{\Delta y}\right]$$

$$+ \frac{(\bar{\eta}H)_B^{i,j}}{\Delta y}\left[\frac{u_{Cx}^{i,j+1} - u_{Cx}^{i,j}}{\Delta y} + \frac{v_{Cy}^{i+1,j} - v_{Cy}^{i,j}}{\Delta x}\right] \quad \text{(A4)}$$

$$- \frac{(\bar{\eta}H)_B^{i,j-1}}{\Delta y}\left[\frac{u_{Cx}^{i,j} - u_{Cx}^{i,j-1}}{\Delta y} + \frac{v_{Cy}^{i+1,j-1} - v_{Cy}^{i,j-1}}{\Delta x}\right]$$

$$- \beta_{eff,Cx}^{i,j} u_{Cx}^{i,j} = -\tau_{d,x,Cx}^{i,j}.$$

Assuming that $\Delta x = \Delta y = \Delta$ (which is the case in IMAU-ICE), multiplying both sides by $\Delta^2$, and defining the product term $N = \bar{\eta}H$, Eq. A4 can be rearranged to read:

$$u_{Cx}^{i,j}\left(-4N_A^{i+1,j} - 4N_A^{i,j} - N_B^{i,j} - N_B^{i,j-1} - \Delta^2\beta_{eff,Cx}^{i,j}\right)$$

$$+ u_{Cx}^{i+1,j}\left(4N_A^{i+1,j}\right) + u_{Cx}^{i-1,j}\left(4N_A^{i,j}\right) + u_{Cx}^{i,j+1}\left(N_B^{i,j}\right) + u_{Cx}^{i,j-1}\left(N_B^{i,j-1}\right)$$

$$+ v_{Cy}^{i,j}\left(-2N_A^{i,j} - N_B^{i,j}\right) + v_{Cy}^{i+1,j}\left(2N_A^{i+1,j} + N_B^{i,j}\right) + v_{Cy}^{i,j}\left(2N_A^{i,j} + N_B^{i,j-1}\right) + v_{Cy}^{i+1,j}\left(-2N_A^{i+1,j} - N_B^{i,j-1}\right) \quad \text{(A5)}$$

$$= -\Delta^2\tau_{d,x,Cx}^{i,j}.$$

Eq. A5, together with the equivalent representation of the second DIVA equation, can be represented by a sparse matrix equation (with 9 non-zero elements per row), which can be solved by any desired matrix solving algorithm (the default in IMAU-ICE is PETSc, though a generic SOR-solver can alternatively be used). The strain rates $u_x, u_y, v_x, v_y$, the effective viscosity $\eta$, and the product term $N$ are all calculated on the regular A-grid; $N_B$ is obtained by staggering $N_A$. The sliding term

$\beta_{eff}$ is calculated on the A-grid, then staggered to the Cx/Cy-grids, where it is scaled with the grounded fraction (which is calculated directly on the Cx/Cy-grids).

## Appendix B – ISMIP-HOM results

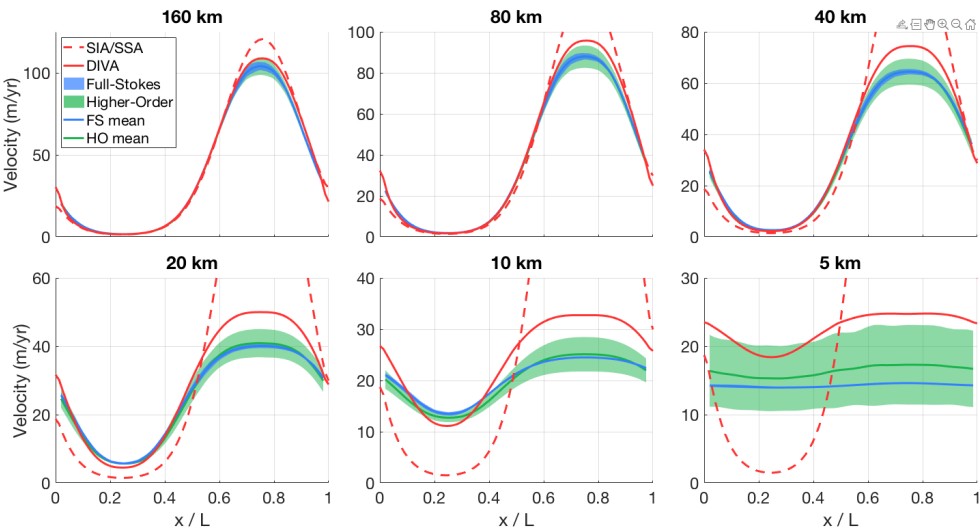

**Figure B1: Modelled surface velocity transects for all versions of ISMIP-HOM experiment A (infinite ice slab on a sloping bed with sinusoid bumps in both directions), calculated with both the old hybrid SIA/SSA solver (red dashed line) and the new DIVA solver (red solid line). The results of the higher-order models (green) and the full-Stokes models (blue) that participated in ISMIP-HOM are shown for comparison.**

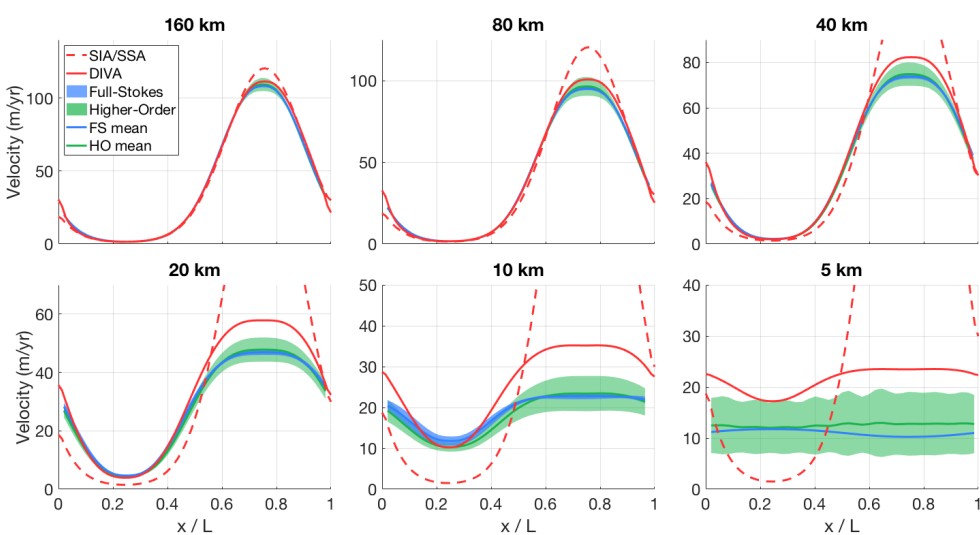

**Figure B2: Modelled surface velocity transects for all versions of ISMIP-HOM experiment B (infinite ice slab on a sloping bed with sinusoid bumps in one direction), calculated with both the old hybrid SIA/SSA solver (red dashed line) and the new DIVA solver (red solid line). The results of the higher-order models (green) and the full-Stokes models (blue) that participated in ISMIP-HOM are shown for comparison.**

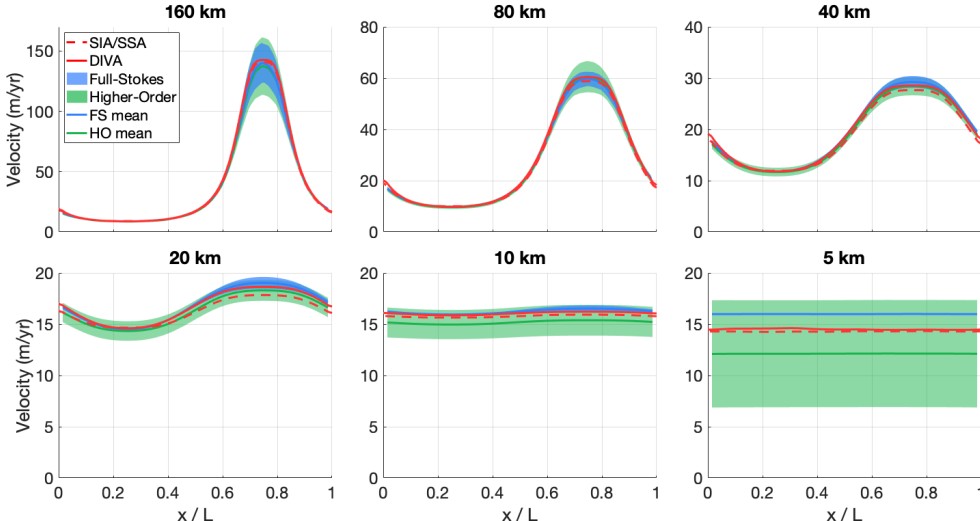

**Figure B3: Modelled surface velocity transects for all versions of ISMIP-HOM experiment C (infinite ice slab on a sloping bed with oscillating friction in both directions), calculated with both the old hybrid SIA/SSA solver (red dashed line) and the new DIVA solver (red solid line). The results of the higher-order models (green) and the full-Stokes models (blue) that participated in ISMIP-HOM are shown for comparison.**

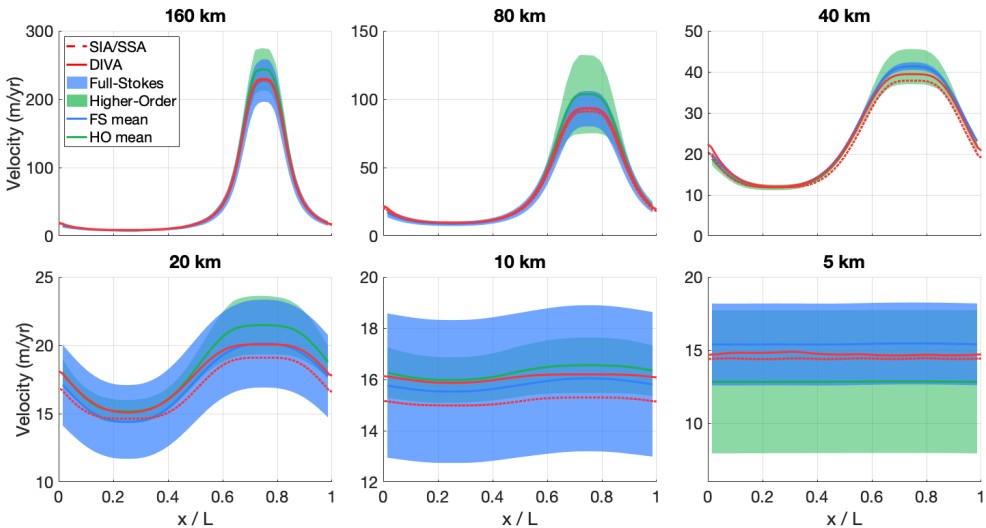

**Figure B4: Modelled surface velocity transects for all versions of ISMIP-HOM experiment D (infinite ice slab on a sloping bed with oscillating friction in one direction), calculated with both the old hybrid SIA/SSA solver (red dashed line) and the new DIVA solver (red solid line). The results of the higher-order models (green) and the full-Stokes models (blue) that participated in ISMIP-HOM are shown for comparison.**

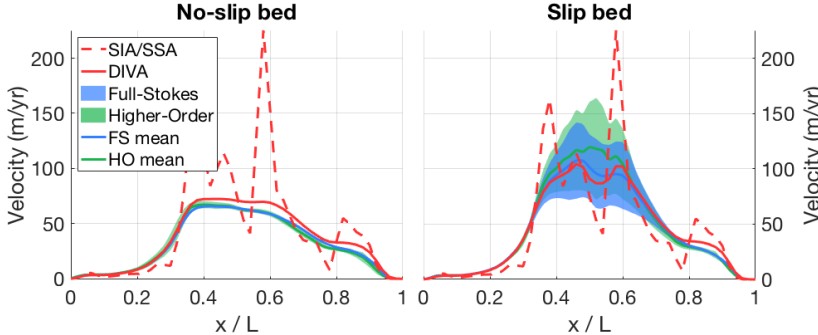

**Figure B5: Modelled surface velocity transects for both versions of ISMIP-HOM experiment E (Haut Glacier d'Arolla, with and without a slippery bed region), calculated with both the old hybrid SIA/SSA solver (red dashed line) and the new DIVA solver (red solid line). The results of the higher-order models (green) and the full-Stokes models (blue) that participated in ISMIP-HOM are shown for comparison.**

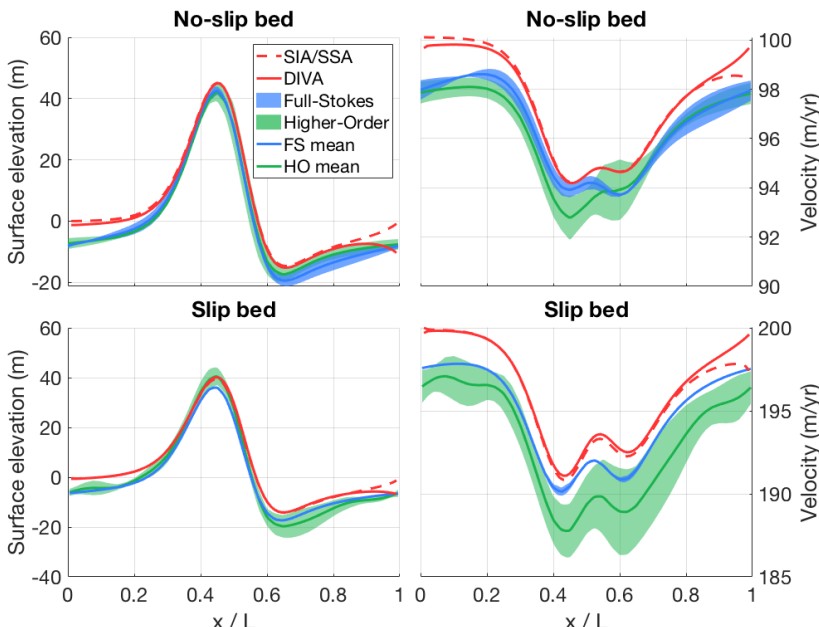

**Figure B6: Modelled surface elevation and velocity transects for both versions of ISMIP-HOM experiment F (infinite ice slab on a sloping bed with a single Gaussian bump, with and without a slippery bed), calculated with both the old hybrid SIA/SSA solver (red dashed line) and the new DIVA solver (red solid line). The results of the higher-order models (green) and the full-Stokes models (blue) that participated in ISMIP-HOM are shown for comparison.**

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
