# Peer review of "Benchmarking the vertically integrated ice-sheet model IMAU-ICE (version 2.0)"

_Geoscientific Model Development, 2021_

## Referee Comment (RC1)

**Review of: Benchmarking the vertically integrated ice-sheet model IMAU-ICE by Berends *et al**

Evan J. Gowan

evangowan@gmail.cm

Berends *et al* present a description of the ice sheet model IMAU-ICE version 2.0. The main feature of this model is the use of the depth-integrated viscosity approximation (DIVA) for solving the stress balance. In the paper, the authors provide a detailed overview of the model, then run a series of standardized tests to demonstrate the performance of the model versus several analytical or higher order solutions. The results show that the DIVA solver provides an improvement of the ice sheet evolution over the more commonly used hybrid shallow ice approximation/shallow shelf approximation model.

Overall, I found the paper was well written and easy to follow. I have a few points for the authors to consider.

**1 Suggestions for the paper**

**1.1 Variable definitions**

There are a number of equations introduced in section 2, and I found that in a few cases the variables were not explicitly defined. I suggest double checking this. It may also be helpful to include a table with all the variables (perhaps even in the appendix).

**1.2 MIPs**

There are a number of tests applied to the ice sheet model. Some of these MIPs I was previously not aware of. Though they are defined in section 4, they are sometimes referenced earlier in the paper. I think it would be helpful somewhere early on to have a table with the different MIPs, what they mean and/or what they are testing, and perhaps the main result of the tests with IMAU-ICE version 2.

**1.3 Figure 3**

It is very difficult to see the results between the different model runs in this figure. Perhaps a better way to display this would be to show it as a difference from Schoof's analytical solution rather than as a raw velocity value.

**2    IMAU-ICE software**

I tried to get the software running using the instructions on the Github repository. Unfortunately, I had a bit of trouble compiling the program. After finally getting it to compile, I was unable to run the test case. I posted an issue about this on Github (`https://github.com/IMAU-paleo/IMAU-ICE/issues/22`). I think it would be worthwhile for the authors to try to get the program running on other systems to ensure ease of use, as it is an explicit goal of this model.

Best Regards,

Evan J. Gowan

---

## Referee Comment (RC3)

**Review of manuscript gmd-2021-352 "Benchmarking the vertically integrated ice-sheet model IMAU-ICE (version 2.0)" by C.J. Berends et al.**

General comments:

The submitted manuscript gives an overall clear and well-structured description of the ice-sheet model IMAU-ICE 2.0, which seems to be a suitable model for running long-term, continental-scale (paleo) simulations. The key developments that led to the release of version 2.0 of the model as well as its main features in general are described in a clear and understandable way. The benchmark experiments performed with the new model version are presented concisely, though in places I am missing a more in-depth discussion of the results (see specific comments below). In places more a few more references would enrich the manuscript.

I would support the publication of the manuscripts after the points below have been addressed.

Sincerely,
Johannes Feldmann

Specific comments:

My main point here is that while the number of benchmark experiments carried out for this paper is very convincing, I wondered why the MISMIP+ benchmark is not part of the analysis. It is the state-of-the-art benchmark regarding grounding-line stability and migration under the influence of strong buttressing. The experiments thus provide insight in how well a model can represent ice-flow dynamics on a smaller spatial scale. There might be good reasons why the authors neglected these experiments but I strongly suggest that the reasons should be at least mentioned in the discussion. Also, the prescribed spatial resolutions vary strongly between the different benchmarks carried out for this study. It would be helpful to give a short reasoning for the chosen resolution ranges. Please see the list below for further specific comments:

P1,L23: Not able to find van de Wal, 2019 in the reference list. Please consider other literature as well.

P1,L27-30, P2,L1-2: Consider adding literature that 1) gives examples of short-term future projections, long-term paleo simulations and 2) relate to the mentioned physical processes

P1,L24-26: This sounds like quite a strong statement to me. I am not sure whether this claim is explicitly supported by the cited study. I would suggest a different wording here. In detail, I don't see from the cited study that the SIA/SSA method has been shown to lead to unsatisfactory results.

P4,L3-4: This point is not entirely clear to me. I can see from Fig. 1 that there are three regions overlapping in the northern hemisphere. But I would wish to have a bit more detail on what is meant by double-counting. Does "no ice growth mean" in the mentioned regions mean that there will be no ice at all or does it mean that already existing ice cannot grow thicker?

P4,L6: Please add information to the figure caption on what the colors show (ocean + bathymetry/bed topography?)

P4,Eq1: I'm missing a brief expalanation of the notation (indices x and y refer to derivatives, bars are vertical averages). Also, the description of the variables is incomplete (e.g. $u$ and $v$)

P4,Sec.2.2: The introduction mentions the advantages of the DIVA approach compared to the hybrid SIA/SSA approach and briefly mentions which stress terms the DIVA approach covers. Sec. 2.2, that includes the mathematical equations of the stress balance would be suited to refer to these stress terms. I suggest to name which of the shown equations/terms correspond to which stress terms (SIA, SSA and additional stresses that are not captured by the SIA/SSA). That would give a lot more clarity on what the actual difference between DIVA and hybrid SIA/SSA is.

P6,L6-7: I would suggest to delete "the square of" for more clarity.

Figure 6: I am surprised that the velocity deviation of the SSA (red-dashed) to the Stokes reference (blue) increases with finer spatial resolution. Is there a plausible explanation for this?

P12,L16: Which version of IMAU-ICE is meant here? Please check also for possible other occurrences where the version is not given but relevant.

P13,L4: I wonder why the authors did not examine finer resolutions. A brief explanation here or in the discussion would be very helpful.

P13,L14: I am not familiar to the Robin solution. For the interested reader, at least a reference should be provided.

P16,L14: I would be interested in more details on the simplicity of the mentioned rheology, damage and subglacial hydrology. I recommend to discuss them here or to present details in the section 2.

Figure 9/10: As the shown results are very similar for ABUM and ABUK, maybe it is sufficient to show only one of the two figures in the main text (shifting the other into the supplement).

Figures B3/B4: There are no red-dashed lines visible in both figures. Does this mean that results from DIVA and SIA/SSA are identical here? If so, it would be good to mention this in the figure caption.

References: The list as it is presented makes it hard to identify the individual studies. It needs vertical spaces between the individual references.

---

## Author Comment (AC1)

**Rebuttal to the review by Evan Gowan.**

We thank the reviewer for their insightful and constructive comments on our manuscript. We'd hereby like to address their concerns, and propose revisions to our manuscript to alleviate them. Reviewer's comments are displayed in boldface, replies in regular type.

**There are a number of equations introduced in section 2, and I found that in a few cases the variables were not explicitly defined. I suggest double checking this. It may also be helpful to include a table with all the variables (perhaps even in the appendix).**

We will extend Table 1 to include definitions of all model variables and parameters.

**There are a number of tests applied to the ice sheet model. Some of these MIPs I was previously not aware of. Though they are defined in section 4, they are sometimes referenced earlier in the paper. I think it would be helpful somewhere early on to have a table with the different MIPs, what they mean and/or what they are testing, and perhaps the main result of the tests with IMAU-ICE version 2.**

We will add a table to the end of the Introduction section listing, and briefly describing the purpose of, the different benchmark experiments.

**It is very difficult to see the results between the different model runs in this figure. Perhaps a better way to display this would be to show it as a difference from Schoof's analytical solution rather than as a raw velocity value.**

Based on the suggestion by the anonymous reviewer, we have changed this experiment. The ice stream is now wider, so that the same resolutions as for the other experiments (40, 32, 20, 16, 10 km) can be used. The modelled results are now clearly visible.

**I tried to get the software running using the instructions on the Github repository. Unfortunately, I had a bit of trouble compiling the program. After finally getting it to compile, I was unable to run the test case. I posted an issue about this on Github (https://github.com/IMAU-paleo/IMAU-ICE/issues/22). I think it would be worthwhile for the authors to try to get the program running on other systems to ensure ease of use, as it is an explicit goal of this model.**

Because of the dependency of the model on external libraries (NetCDF, Lapack, PETSc), which are not located in the same place on every system, we cannot guarantee that it will run out-of-the-box on any platform. We will have another look at the predefined config files on Github to make sure those all work once the model is successfully compiled.

---

## Author Comment (AC2)

**Rebuttal to the review by an anonymous reviewer**

We thank the reviewer for their insightful and constructive comments on our manuscript. We'd hereby like to address their concerns, and propose revisions to our manuscript to alleviate them. Reviewer's comments are displayed in boldface, our replies in regular type.

Please note that all comments are answered in the original order, with one exception; since the reviewer's comments about the sub-grid melt scheme, the sliding law, and the ABUMIP results are strongly related, we answer those together.

**Major comments**

**One goal of this paper is to convince the reader that the model can be used for scientific studies using a resolution of 16km. While many studies have shown this is too coarse to accurately represent ice sheet dynamic accurately, it can be a good trade off between computing cost and model accuracy especially for paleo time scale simulations. Keeping this in mind, I would have liked this paper focusing more on convincing the reader it could do so and quantify the errors with higher resolution results in the experiments presented here. In many of the experiments, either 16km is the highest resolution, or higher resolutions is used without comparing with results at 16km. For this reason, I find this study incomplete, and I strongly encourage the authors to add simulations to their suite of runs. Also, please, be a bit more quantitative in your analysis of each experiment.**

The main goal of our paper is to convince the reader that the model can be used for palaeoglaciological applications. We typically use a resolution of 40 km for this kind of work, because of the very long timescales involved; while 16 km might be regarded as "coarse" within the context of future projections, for palaeo-applications this is generally already too high to be feasible. By showing that, in the different benchmark experiments, our model results at 40 km do not appreciably deviate from those at higher resolutions, we demonstrate that this relatively coarse resolution still produces reliable results.
We will clarify the text in the introduction to more clearly reflect the fact that 40 km should be viewed as the default, and that the higher-resolution experiments serve only to illustrate that 40 km does not introduce prohibitively large truncation errors. Having said that, the suite of experiments also show that the model can in principle be used at a higher resolution.

**I am a bit concerned about your need to scale your basal stress by the square of the grounded area fraction as opposed to its real value. It reduces the amount of basal stress applied in the cell containing the grounding line, and in some cases, by a lot (by half when the grounding line is at the middle of that cell). I am not sure why and it might be due to the implementation of your numerics, or else. For area faction less than 0.5, it will behave closely to a model not using any grounding line parameterization which will lead to reduced model accuracy. Have you tried using the same logic used for the basal stress parameterization as the one used for your basal melt rate parameterization (i.e., considering the cell grounded if the cell center is grounded and floating otherwise)? Also, you applied this scaling using numerical justification, do you have any physical justification you could add to it?**

We based our approach to the sub-grid friction scaling largely on that used in PISM, as described by Feldmann et al., 2014. In that paper, they describe not only a sub-grid scaling scheme, but also how the driving stress is calculated using two-sided differencing everywhere except for near the grounding line, where one-sided differencing is used. In practice, this introduces the same quadratic dependence on grounded area that we use. The difference between the two approaches has to do with the choice of staggered grids. PISM solves the SSA on the regular grid, so that the driving stress can be calculated using either two-sided or one-sided differencing. In IMAU-ICE, the SSA and the DIVA are both solved on the staggered grid, where one-sided differencing is a bit more problematic (because it would involve information from non-nearest-neighbour grid cells). We believe that scaling the basal friction with the square of the sub-grid grounded fraction is therefore simply a slightly different way to achieve the same result. Our preliminary experiments, where we did not include the square, showed similar results to what Feldmann et al. describe when they do not include the one-sided differencing scheme, which further strengthens this belief.
We will clarify the text of the manuscript to reflect these thoughts.

**About your MISMIP experiments: the advantage of this experimental setup is that it has an analytical solution. It would be good to see how your results compare to it. Pattyn (2017) uses a flux condition at the grounding line that is derived from the boundary layer solution from Schoof (2007), hence his spun up steady state is really accurate despite the use of a coarse resolution. It looks like your results are still about 100km away from the analytical solution at the end of the original spun up state with your 16km (highest) resolution. Also, at the end of the re-advance experiment, the difference in grounding line location between the 32km and the 16km resolutions results is much bigger than the difference between the 64km and the 32km resolutions results, and these differences seem to be about the same at the end of the retreat experiment. It would be sensible for you to show results at higher resolutions (8km and 4km for example) for this experiment. This will help quantify the error with 16km.**

The experiment we performed follows that of Pattyn (2017), where the original MISMIP experiment is modified from a 1-D flowline to a 2-D plan-view setting more appropriate for continental ice-sheet models. This means that the analytical solution by Schoof (2007), which was derived for a flowline case, is no longer applicable. We will clarify this difference in the manuscript to avoid confusion.
We will also perform a few additional simulations so that both the (modified) MISMIP and the ABUMIP experiments are presented at the same 40, 32, 20, 16, and 10 km resolutions. We will furthermore add a panel to the MISMIP figure showing the relation between grid resolution and GL hysteresis, which clearly shows how the amount of hysteresis decreases with resolution, and is always smaller than the grid resolution (which we believe is the most important result of this experiment).
We also fixed a small error in the plotting script that creates this figure, so that the GL position curves look smoother now (before, it calculated the position from the discrete flotation mask, this has now been changed to utilise the sub-grid GL position).

As you mention in your introduction, only a few existing ice sheet models are using DIVA. Thus, the best comparison possible to other models would be to employ similar experimental setup they have used too. You cite the work from Leguy et al. 2021 which has the advantage to present results with several sliding laws (including coulomb friction) and the same melt parameterization you chose for your model (on top of using DIVA as well). IT would have been sensible to implement some of their experimental setup and present results from your model. Especially since you are using their study to justify your choice of melt parameterization. I will get back to this below.

About your ABUMIP experiments: you are using ABUMIP to test your melt parameterization. While this experiment is an application of using your basal melt rate parameterization, it is not designed to demonstrate how well it works in your model. In this experiment, the melt rate is large and Leguy et al. (2021) showed that the melt rate parameterization is very resolution sensitive under these circumstances. Your model barely shows sensitivity with resolution. Also, IMAU-ice is the only model using a coulomb friction law preventing you to make a sensible comparison with other models in the study. Right now, you are comparing 2 different version of your own model and your results are even more extreme compared to the previous version. What do you make of this? Which results would you believe is the most sensible? (I am not saying it is wrong but comparing one outlayer with another just takes you so far.) A more meaningful comparison would have been to run with a Weertman sliding law and compare your results with CISM which is the only model using DIVA in Sun et al. (2020) (listed as L1L2). Physically speaking, you should expect ABUK to produce more sea level rise than ABUM. Eventually both experiments should plateau towards the same results but at the least, ABUM should be lagging behind ABUK. I am concerned that all your runs for a given resolution are similar for both ABUM and ABUK, and your results at 16km are the one leading to the most sea level rise. Also, you can see that your run at 40km returns more sea level rise with ABUM than with ABUK. You should discuss this more in your text and discard the results at that resolution for this reason. (A few models in Sun et al. (2020) exhibit this behavior and I don't think it is right either.) Again, showing performance with the setup from Leguy et al. (2021) or Seroussi et al. (2018) would help explain a lot of these behaviors. I do think adding this experiment is adequate for this paper and does not need to wait for another publication especially since you are claiming good results with your newly implemented melt parameterization.

I am concerned about your first paragraph in your future research section which I highlights here in 3 points:
1) The study from Leguy et al. (2021) clearly highlights that the choice of basal melt parameterization is model dependent. In fact, using a similar melt parameterization as in Seroussi et al. (2018), they found opposite results. This means that you should test your model first and convince yourself and reviewers which conclusion you model reaches.
2) You mention a factor of 2 of sea level rise differences depending on the choice of melt parameterization; was it twice as much or half of the presented results? If the former, that is indeed scary. If the later, why would you think it is not the right choice to make? Leguy et al. (2021) found that using PMP or FCMP lead to similar results.

3) **You state that the NMP and PMP schemes were resolution dependent, inferring that the FCMP scheme isn't, which would explain your ABUK and ABUM results. Leguy et al. (2021) shows that all melt parameterizations are grid resolution dependent for high melt experiments, even more so when using a coulomb friction law. (This was true as well in Seroussi et al. (2018) in some cases and their coarsest resolution was 2km.) I would expect this to be true in your model as well.**

We agree that the interplay between the sliding law, the sub-grid melt scheme, and the modelled grounding-line retreat deserve to be investigated more thoroughly, and that the schematic MISMIP+-type of experiments performed by Leguy et al. (2021) and the more realistic ABUMIP-type of experiments provide a solid basis for this. In fact, we have already performed a large number of these experiments over the past few months, as part of the basal sliding / basal melt upgrade mentioned in the Discussion section of our manuscript. Regarding the relative "performance" of the sub-grid melt schemes, our findings match those reported by Leguy et al., meaning that it depends on the experimental set-up (i.e. schematic MISMIP+ v.s. realistic Antarctica) and on the model resolution. Likewise, we find different results in the MISMIP+ vs the Antarctica set-ups regarding the effect of sliding laws; in MISMIP+, the effect of choosing a different sliding law is larger than that of choosing a different melt scheme, whereas in Antarctica we find the opposite. Regarding the comparison to other, similar models: we have also performed all of the MISMIP+ experiments (again with different resolutions, sliding laws, sub-grid schemes, and also different stress balance approximations), and find that all of our simulations produce grounding-line retreat curves that lie well within the range of other models (as reported by Cornford et al., 2020).

While we agree that these are certainly valuable experiments, and we definitely want to publish them, we feel that they are of a different nature than the work presented in the current manuscript. Whereas the current work consists of more basic numerical verification experiments, these new experiments move to a next level, studying the effects of different physical parameterisations on the rate of change of the Antarctic ice sheet. A discussion of this breadth and depth would require more text than could be feasibly added to the current manuscript, which is already almost 30 pages long. We therefore decided to move them to a separate publication, where we can leave out a lot of model description and benchmark experiments (as they will already be in the current manuscript), and spend the available text on a more thorough investigation of these model choices. We will mention these ongoing developments more extensively in the "future research" part of the Discussion section.

Regarding the results for the ABUM/ABUK experiments: the extremely high melt rate of 400 m/yr that is prescribed in the ABUM experiment should lead to a lag of 3 or 4 years at the most with respect to the ABUK results; in the 500-year figures used here and by Sun et al. (2020), this difference would not be visible. In the extreme case of a grounding-line thickness and velocity of 2,000 m and 1,000 m/yr, respectively, this would lead to a shelf of only 2.5 km long, which we doubt would generate enough buttressing to cause any significant differences between ABUM and ABUK. We will reflect these thoughts in the manuscript.

**Minor comments**

**P2, l32: your reference to Bueler and Brown (2009) is not listed in your references. Please add it.**

We will do so.

**P4, eq2: Please define your integration bounds "b" and "s", and your exponent "n". I note that you define "b" later on page 5 and it is always nicer for the reader to know about the different notations as they see them.**

Following a suggestion by reviewer #1, we will include a table listing all model symbols and parameters.

**P5, l9: please explicitly write down your Neuman boundary condition.**

We will do so.

**P5, l17: please define z_sl.**

This will be included in the symbol table.

**P6, l1: You mention that \lambda_w is limited between 0 and 1. As written on P5, l18 it is not bounded to do so. I would suggest to write it as: \lambda_w = min(z_sl – b, 1000), when b < z_sl, or something similar. If your code does not do that, please modify your text accordingly. Similar remark for w_b and equation 10.**

We will add the 0 – 1 limits to the equations.

**P3, Sec2: what kind of horizontal grid discretization are you using? On P6, l18 you mention a staggered grid without talking about what kind of staggering you are using. Also, refer to the appendix for more details about it.**

A reference to Appendix A was indeed missing from Section 2; we will include it.

**P7, l29-30: please write down the equation you are using for basal mass balance for clarity since you are combining 2 methods. Plus, it reduces the amount of paper the reader as to go through.**

We will do so.

**P9, l3: replace Sec3.4 by Sec 3.2. At least in the printable version, there are no sections 3.2 and 3.3.**

We will do so.

**P9, Sec.3.4: why did you choose your coarsest resolution to be 5km here while you test much coarser resolution in all the other experiments? See remark above. (You could simply show results at 0.5km, 1km, 2km, 4km, 8km, 16km, 32km.)**

We initially chose the same set-up as Bueler and Brown (2009), which describes a narrow (~80 km) ice stream. We agree that it is more meaningful to choose a set-up where the same resolutions as for the other experiments (40, 32, 20, 16, 10 km) can be used. We have redone the experiments with a new set-up, describing a wider (~300 km) ice stream, so that a 40 km resolution can still resolve the low-friction channel. We will update the manuscript to reflect these changes.

**P12, Sec.4.3: Please be more quantitative about the grounding line position at the different resolution and at the different stage of the experiment. You could list them in a table. Also, I am not sure I understand the reference to Leguy et al. (2021) in this section as they use the MISMIP3d (Pattyn et al. 2013) and MISMIP+ experiments (Asay-Davis et al. 2016) in their paper, not MISMIP as opposed to the Leguy et al. (2014) paper. Also showing results at higher resolution would be sensible.**

The second panel we will add to Fig. 7, showing the magnitude of the grounding-line hysteresis as a function of the model resolution, will help clear this up.

The reference to Leguy et al. (2021) should have been to Leguy et al. (2014), which is the study looking at MISMIP.

The new experiments we have done now go as high as 10 km.

**P13, Sec.4.4: Please show a sea level time series of your control experiment.**

We will replace Figs. 8 and 9 with a single figure showing ABUC (the control experiment), ABUM, and ABUK. We will also add a few lines to the text to discuss the results of this experiment.

**P13, l12: I suggest removing "Before starting the 500-year simulation" and begin your sentence directly with "We initialize". It could be confusing especially since you are using a relaxation time of 500 year in your spinup procedure which is not part of the simulation.**

We will do so.

**Fig2: please indicate what O(R^1.1) represents and what R is**

R is the grid resolution; O(R^1.1) means that the error in the modelled margin position scales with the grid resolution to the 1.1'th power (i.e. approximately linearly). We will clarify this in the figure caption.

**Fig 8. And 9.: the results from Sun et al. (2020) show the difference w.r.t to the control experiment. Your figure caption does not state this. Can you clarify whether you subtracted it or not?**

Sun et al. (2020) show the sea-level change with respect to the initial (i.e. spun-up) state, not to the sea-level change in the control run. We do the same.

---

## Author Comment (AC3)

**Rebuttal to the review by an anonymous reviewer**

We thank the reviewer for their insightful and constructive comments on our manuscript. We'd hereby like to address their concerns, and propose revisions to our manuscript to alleviate them. Reviewer's comments are displayed in boldface, replies in regular type.

Please note that all comments are answered in order, with one exception; since the reviewers comments about the sub-grid melt scheme, the sliding law, and the ABUMIP results are strongly related, we answer those together.

**Major comments**

**My main point here is that while the number of benchmark experiments carried out for this paper is very convincing, I wondered why the MISMIP+ benchmark is not part of the analysis. It is the state-of-the-art benchmark regarding grounding- line stability and migration under the influence of strong buttressing. The experiments thus provide insight in how well a model can represent ice-flow dynamics on a smaller spatial scale. There might be good reasons why the authors neglected these experiments but I strongly suggest that the reasons should be at least mentioned in the discussion.**

This is a good point, and one that was also raised by anonymous referee #2. We fully agree with both reviewers that the MISMIP+ experiment is very valuable. We are currently finishing up a project where we investigate the interplay between different sliding laws, basal melt parameterisations, sub-grid melt schemes, stress-balance approximations, and grid resolutions. We did this by performing two large ensembles of simulations; one with the schematic MISMIP+ geometry, and one with the present-day Antarctic geometry.

While we definitely want to publish these experiments, we feel that they are of a different nature than the work presented in the current manuscript. Whereas the current work consists of more basic numerical verification experiments, these new experiments move to a next level, studying the effects of different physical parameterisations on the rate of change of the Antarctic ice sheet. A discussion of this breadth and depth would require more text than could be feasibly added to the current manuscript, which is already almost 30 pages long. We therefore decided to move them to a separate publication, where we can leave out a lot of model description and benchmark experiments (as they will already be in the current manuscript), and spend the available text on a more thorough investigation of these model choices. We will mention these ongoing developments more extensively in the "future research" part of the Discussion section.

We here include a figure from the manuscript of this new project, showing the results of the default MISMIP+ simulations, compared to the ensemble results published by Cornford et al. (2020). As we see, the results of IMAU-ICE lie well within the ensemble range, indicating that the model performs well.

[Figure]

Figure 1: grounding-line position over time for the different MISMIP+ experiments. Solid lines (shaded areas) depict the mean (range) of results for IMAU-ICE. Dotted (dashed) lines depict the mean (range) of results for the model ensemble from Cornford et al. (2020). Colours indicate the ice0 (unforced control), ice1r (100-yr retreat), ice1ra (100-yr readvance), and ice1rr (100-yr continued retreat) experiments (see Cornford et al. (2020) or Asay-Davis et al. (2016) for a more detailed explanation). Panel A shows the ice1 experiments (fixed calving front, variable melt rate), panel B shows ice2 (high melt near the calving front).

**Also, the prescribed spatial resolutions vary strongly between the different benchmarks carried out for this study. It would be helpful to give a short reasoning for the chosen resolution ranges**.

**P13,L4: I wonder why the authors did not examine finer resolutions. A brief explanation here or in the discussion would be very helpful.**

We agree that the choice of resolutions should be more consistent across the experiments. We have redone all of the simulations for the plan-view MISMIP, SSA ice stream, and ABUMIP experiments, at resolutions of 40, 32, 20, 16, and 10 km. The results remain qualitatively unchanged. This range of resolutions is motivated by the intended application of IMAU-ICE to palaeoglaciological experiments at a 40 km resolution. By showing that the model results remain unchanged in a variety of settings even at resolutions as high as 10 km, we provide confidence that the 40 km results are still reliable.
We will adapt the figures and the manuscript text to reflect these changes.

**P1,L23: Not able to find van de Wal, 2019 in the reference list. Please consider other literature as well.**

van de Wal et al. (2019) is already in the list of references.

**P1,L27-30, P2,L1-2: Consider adding literature that 1) gives examples of short- term future projections, long-term paleo simulations and 2) relate to the mentioned physical processes.**

For the short-term future projections, we will add references to Goelzer et al. (2020), Levermann et al. (2020), Seroussi et al. (2020), and Sun et al. (2020). For the long-term paleo simulations, we will add Abe-Ouchi et al. (2013), Berends et al. (2018, 2019, 2021), de Boer et al. (2013), and Willeit et al. (2019). For the physical processes, we will add de Boer et al. (2014) for GIA, Berends et al. (2018, 2019) for feedbacks of ice-sheet geometry on the regional, and Abe-Ouchi et al., 2013 for changes in orbital configuration.

**P1,L24-26: This sounds like quite a strong statement to me. I am not sure whether this claim is explicitly supported by the cited study. I would suggest a different wording here. In detail, I don't see from the cited study that the SIA/SSA method has been shown to lead to unsatisfactory results.**

This reference is indeed incorrect; no SIA/SSA results were included in the original ISMIP-HOM study. We will instead refer to Goldberg (2011), who did make the comparison we were thinking of.

**P4,L3-4: This point is not entirely clear to me. I can see from Fig. 1 that there are three regions overlapping in the northern hemisphere. But I would wish to have a bit more detail on what is meant by double-counting. Does "no ice growth mean" in the mentioned regions mean that there will be no ice at all or does it mean that already existing ice cannot grow thicker?**

It means no ice at all. For example, the North America region of the model contain a permanently ice-free Greenlandic island. We will clarify this in the manuscript.

**P4,L6: Please add information to the figure caption on what the colors show (ocean + bathymetry/bed topography?)**

We will do so.

**P4,Eq1: I'm missing a brief expalanation of the notation (indices x and y refer to derivatives, bars are vertical averages). Also, the description of the variables is incomplete (e.g. u and v).**

We will add a table listing the model symbols, and we will clarify the notation.

**P4,Sec.2.2: The introduction mentions the advantages of the DIVA approach compared to the hybrid SIA/SSA approach and briefly mentions which stress terms the DIVA approach covers. Sec. 2.2, that includes the mathematical equations of the stress balance would be suited to refer to these stress terms. I suggest to name which of the shown equations/terms correspond to which stress terms (SIA, SSA and additional stresses that are not captured by the SIA/SSA). That would give a lot more clarity on what the actual difference between DIVA and hybrid SIA/SSA is.**

We will add these clarifications to the text immediately after Eq. 1.

**P6,L6-7: I would suggest to delete "the square of" for more clarity.**

We will not do this, as the response by anonymous referee #2 indicates that this is an important detail.

**Figure 6: I am surprised that the velocity deviation of the SSA (red-dashed) to the Stokes reference (blue) increases with finer spatial resolution. Is there a plausible explanation for this?**

The distances listed above the figure panels do not refer to model resolution, but to the spatial scale of the bumps in the bedrock (although the model resolution also changes between the experiments, to maintain a similar relative truncation error). The geometry of the experiment describes sinusoidal bumps superimposed on a sloping plane; the numbers refer to the wavelength of the bumps. Experiment A therefore implies a high aspect ratio (1 km of ice, 150 km wavelength in the bedrock bumps), so that both the hybrid SIA/SSA and the DIVA are reasonably accurate. In each subsequent experiment the aspect ratio decreases, making both approximations increasingly inaccurate, regardless of model resolution. We will clarify this in the manuscript.

**P12,L16: Which version of IMAU-ICE is meant here? Please check also for possible other occurrences where the version is not given but relevant.**

All experiments described in the manuscript are performed with IMAU-ICE v2.0, which is the model version being presented here. Only in the ABUMIP experiment do we compare to the previously published results of version 1.0, which is stated explicitly in the text.

**P13,L14: I am not familiar to the Robin solution. For the interested reader, at least a reference should be provided.**

It is an analytical solution for the temperature profile in an ice clumn, given a profile of the vertical velocity, thermal parameters, and geothermal heat flux. We will clarify this in the text, and add a reference to Robin (1955).

**P16,L14: I would be interested in more details on the simplicity of the mentioned rheology, damage and subglacial hydrology. I recommend to discuss them here or to present details in the section 2.**

We will briefly mention the treatment of rheology and damage in section 2. The (lack of) treatment of subglacial hydrology is already included in the description of basal sliding; pore water pressure is calculated solely based on bedrock elevation, following Martin et al. (2011).

**Figure 9/10: As the shown results are very similar for ABUM and ABUK, maybe it is sufficient to show only one of the two figures in the main text (shifting the other into the supplement)**

Taking also into account the comments by the other referees, we will combine these two Figures into a single one.

**Figures B3/B4: There are no red-dashed lines visible in both figures. Does this mean that results from DIVA and SIA/SSA are identical here? If so, it would be good to mention this in the figure caption**

The SIA/SSA results had been accidentally left out of these two figures. Their results are indeed very similar to those by the DIVA (and both are close to the full-Stokes solution, since these experiments all have the same aspect ratio; the perturbation is in the bed roughness instead). We will update the figures.

**References: The list as it is presented makes it hard to identify the individual studies. It needs vertical spaces between the individual references.**

The make-up of the reference list follows the Copernicus manuscript template. In our experience this will become easier to read after typesetting.

---

## Referee Report (RR1)

**Review of Constantijn et al. 2021, GMD (round 2)**

I would like to thank the authors for taking the time to respond to my and other reviewers' comments. I find that the manuscript has improved especially with the presentation of the experiments with the suggested resolutions.

Unfortunately, despite the additional information added to the manuscript, I cannot recommend this manuscript for publication at this time. Rather I would say that if my major concern is not addressed, I would have to reject it (I'll explain a bit more below). Maybe my first round of review was not clear enough, and I apologize about that.

I will start here with a few general points before stating my main concern and ending with other comments.

I feel a bit uneasy with a current tendency of ice sheet modeling papers stating it is fine to use coarse resolution (here 40km) and allow the model to have greater error because it will be used for paleoclimate simulations. I respectfully disagree with this mindset, and the computational expense of long paleoclimate simulations should not be an excuse to (gravely) misrepresent ice sheet behavior. In this paper you also mention that you plan on using the model for future projection study as well (using 16km resolution) meaning this "benchmark" paper should also be used to convince the reader it is acceptable to do so with this model, which it isn't.

In my first round of review, I did request for the sections presenting the experiments to be a bit more quantitative in their discussions; sometimes adding tables is also a good idea. I appreciate an effort was made in adding convergence figures (e.g., figure 7b) and I would like to see more discussion on the relative errors whether with respect to the analytical solutions or with the highest resolution used in these experiments. Simply stating something like "look, we do see convergence with resolution" is over simplistic. In Sec3.2, Fig.3, it is not only difficult to get a sense of the error with respect to the analytical solution (the y axis is not precise enough) but also you state (in your discussion) that your error at 40km is a reliable result for paleoclimate studies. Two arguments here: 1. What is an acceptable/reliable error? You never define this concept (and it is hard to do actually). 2. You seem to argue that an error of 350m/y with respect to the analytical solution is acceptable. I would beg to differ! And this is where a more detailed discussion either here or in the discussion needs to take place explaining why this magnitude of error is good enough for paleoclimate simulations. Based on the results on this section, I would feel way more comfortable if you chose to use a resolution of 20km.

**My major concern**:
In this manuscript, you are showing some numerical capabilities about your model which is one side of the story. The other side of the story is you try to convince the reader that your model is suitable for continental scale simulations as well. (You do show the different domains of possible computation in Fig.1 after all.) So far, for this latter point, you are convincing me that

your model is not ready for continental scale simulations, specifically with your ABUMIP experiments. In our first round of comment, me and another reviewer did recommend you adding some MISMIP+ experiments to show the robustness of your choice of resolution and melt parameterization especially for marine ice sheet configuration. You chose not to do so arguing that it is material for another publication. Doing so puts you at a risk that readers will not believe that your choice of model configuration for ABUMIP is suitable. It does so with me. You mentioned the length of your manuscript for not showing the MISMIP+ experiments here. I will say that this should never be the sole excuse for not adding a scientific result to a manuscript. The gmd journal is actually a good place where I can expect papers to be longer because authors are trying to show development of their models and convince the reader of their scientific capabilities (just what you are trying to do).

I thank you for adding the control experiment to the ABUMIP set of plots, it is very informative. I am concerned you thought it would be good enough to show a control that is drifting without forcing by 1.2 m of sea level rise in 500 years. At the end of that time, both Ronne and Ross ice shelves are almost gone. This is worrisome. And the linear trend is not plateauing based on your figure.

**Before this manuscript can be published**: please perform the Antarctic spin-ups at the different resolutions you are presenting that are long enough to prevent your Antarctic ice sheet to drift so much during the control experiment. Doing so would give the reader confidence you can set up your model for paleo or future Antarctic simulations. In addition, please complete your analysis by showing a difference of your end of spin-up runs with your thickness target for your run at 40km and 16km resolutions (since you plan on using both resolutions for Antarctica for paleo and future runs respectfully). Redo the ABUM and ABUK simulations starting from these new spin-ups. Redo Figure 8 accordingly.

Now that I have witnessed your way of performing Antarctic spin-up**, I would highly encourage** you to show the result of a Greenland spin-up at 20km resolution, since you plan on using this resolution for your future studies. (I am simply asking about a spin-up here, not a transient run of any kind). Such a result would strengthen the proof of concept of your new numerical capabilities (I am thinking about DIVA here).

I understand you have a paper underway (hopefully) about MISMIP+ experiments to complement this one. For the time being,  please add a quick highlight of your MISMIP+ experiment results supporting your default of using the FCMP parameterization from Leguy et al. (2021). (Unless you want to add these experiments to this paper to show your results at 40km down to 10km).

Right now, you are showing that IMAU-ICE is not ready to perform sensible Antarctic simulations. And if you cannot produce a good initial steady state for your ABUMIP experiment then you should take out the ABUMIP experiment from this paper and replace it with something else and revise your text accordingly.

**Other comments**

**P4, l16**: in this paper you have not shown any results of any of the continental ice sheets you plan on running with except one for Antarctica (which is quite unfortunate). Showing an initial thickness from ice sheets you experimented with could add great value to your manuscript. In addition, you specifically refer to future Greenland ice sheet (GrIS) simulations here which brings me to one of my general remarks (see above) that showing a steady state simulation of the GrIS would be a great addition to this manuscript.

**P7, l5**: I suggest rewriting "The way the stress balance is discretize" by "The stress balance discretization".

**P7, eq10 and 12**: my previous comment on these equations might have been misunderstood. I appreciate you adding the bounds 0<=w_b<=1 and 0<=\lambda_w<=1 within the equations for clarity but it is still confusing. What I would like to see specifically written in the text is:
1. that for equation 10, b is bounded between b_min and b_max and you achieve this by writing something like:

$$w\_b = \frac{1}{b_{max} - b\_min} \begin{cases} \max(0, b - b_{min},), \mathrm{b} < 0 \\ \min(b_{max} - b_{min}, \mathrm{b} - b_{min},), \mathrm{b} > 0 \end{cases}$$

2. that d_w is bounded between 0 and 1000 so that the bounds for d_w are satisfied. Right now, you write d_w = z_SL -b. So if z_SL=0 (say see level reference set to today's value) and b=-1100, then d_w=1100 and lambda_w=1.1. So clearly, you are taking the minimum between 1 and 1.1 here. Conversely, if b is above sea level, d_w<0 and in this case, you are taking the maximum between d_w and 0. So please, for clarity write d_w using something like:

$$d\_w = \begin{cases} \max(b_{max}, z_{SL} - b), \mathrm{b} > 0 \\ \min(-b_{min}, z_{SL} - b), \mathrm{b} < 0 \end{cases}$$

3. After defining your equations, in the text (and you deleted it), mention that both w_b and d_w are capped between 0 and 1.

**P10, Table 3**: None of the parameters listed in this table are defined in the text. Please do so either in the table and/or in the text.

**P10, l13**: you deleted the sentence stating the possible melt parameterization options available in IMAU-ICE v2.0. I believe it is good to keep them in the text and stating the default configuration for your version of the model. In the previous version you did mention these options but did not discuss the effect of them on your model output. If you do have results comparing them in the context of Antarctic simulation, please, add this discussion!

**P12, l3**: Can you be a bit more quantitative?  Also, the convergence with resolution seems to be superlinear (at least at the center of the domain) and there is a net gain in using a resolution of

20km as opposed to 40km in these simulations. Why do you think the error at 40km resolution is acceptable? (See previous comments.) (I believe the new figure 3 is more informative compared to the older one; we learn something more about the model itself. Thank you for redoing it. )

**P12, Table 4**: please, align your table headers (Parameter, description,…) on one line.

**P16, l8**: Please rephrase this sentence. The ice flow factor is decreased (increased) as a step function after which you run your model forward in time for 15 kyr (not 25 kyr based on Fig. 7A) to a steady state. Please, add the values for the ice flow factors in your text.

**P17, l6**: This argument alone is insufficient to justify an acceptable result. (If I run an ice sheet model with a resolution of 200km and see an error of 195km, does it make it acceptable to use such a resolution?) Also, this part of your sentence is a repetition of what you already said 2 sentences prior. At 40km, the hysteresis corresponds to about 25% of the grounding line displacement. Here you are benchmarking your model meaning you will refer to it to explain results of future science experiments. I would advise to acknowledge this large error, and in the discussion, give an example of a situation for coupled climate models for which this error could be of small importance.
Deeming an error to be acceptable is a tricky business and at the end modelers will run with whatever they feel comfortable with to justify their science.
What I can see in these experiments is that, again, running with a resolution of 20km leads to well improved results compared to 40 with again a superlinear convergence. Any idea what is happening with your 16km resolution results? This might be of importance since you plan on using this resolution to run continental scale future scenarios. Based on this experiment, it seems that 20km is better suited.

**P17, l14**: why did you choose a spin-up of only 500 yr? Clearly you have not reached a steady state. Please continue your spin-ups until you do so. (See main comments from earlier.)

**P18, Fig.8**: The caption indicates that the results from IMAU-ICE v1.0 are shown by dashed black line. I can only see a plain line. Also, which resolution are you displaying on the left? (And thank you for redoing this figure, it is much easier to read and more informative compared to the previous version.)

**P18, l5**: Yes, your drift in ABUC is quite large for a steady state, especially at 40km. This alone indicates that your simulation is not ready. See earlier comments.

**P18, l7**: I disagree, this should not be improved in future work and should be investigated now. This paper is a benchmark for your future scientific studies, and for this reason it is reasonable to expect that you can perform and show one example of an "acceptable" spin-up. (IMAU-ICE v1.0 could do so according to their ABUC results.) All along you are claiming that you can, and it is reasonable to perform simulations at 40km. Clearly, with this result, it isn't! Arguably, it is not

as well at 20km resolution. After 500 year of control experiments, both the Ross and Ronne ice shelves are already almost gone!

**P19, l4**: Why should this be as expected? I don't think so. Here you are comparing your results with ELMER/ice for the ABUM experiment. In the next sentence you argue that the difference between ABUM and ABUK should be small for a given model within 3-4 years of transient simulation (I'll get back to this later). The ELMER/ice result for ABUK is almost 8m lower than the one of ABUM after 500 years. Why do you think there ABUM results is correct and their ABUK result is wrong? I would argue the opposite!
I am not saying your results are wrong per se, I am saying that your argument for comparison is not sound. Your model configurations are different compared to all the models in Sun et al. (2020). And this is where your melt parameterization analysis could help you out.

**P19, l6-l10**: Do you have any evidence to present (experiments, calculations, ...) to argue a lag of 3-4 years at the most between ABUM and ABUK? If not, I would stay away from it. The one thing you know is that the sea level rise results for ABUK should be greater than the one from ABUM.

**P19, l27**: Some of your results are close to analytical solutions but they do not match! Also, you only compared to ELMER/ice and otherwise your ABUMIP results are outliers compared to other ice sheet models. Please rephrase.

**P19, l32**: I strongly disagree with this statement, see previous comments. Define "reliable results"? This section is perhaps a good place to spend time on arguing why you think this resolution is adequate and cite references to back up your statements. Just saying it does not make it true.

**P20, l6**: replace "With a minimum of effort" by "With minimum effort".

**P20, l7**: a space is missing after ")".

---

## Referee Report (RR2)

**Second review of manuscript gmd-2021-352 "Benchmarking the vertically integrated ice-sheet model IMAU-ICE (version 2.0)" by C.J. Berends et al.**

There are still a few minor comments from my side:

P4,Eq.(1): The meaning of the x and y indices and the bar above the variables is still not given in the text. Please add.

P6,L6-8: I suggested to remove "the square of" because otherwise from that sentenced one would deduce that Feldmann et al, 2014 and Leguy et al, 2021 also use the the square of the grounded fraction. Just wanted to point to this again but would leave it to the authors to change the statement or not.

P17,L15: The authors announced but missed to include the reference Robin (1955).

P21,L9: investigating → investigation

Points from my first review that where announced to be addressed by authors but apparently were not:

My point:
*P4,Sec.2.2: The introduction mentions the advantages of the DIVA approach compared to the hybrid SIA/SSA approach and briefly mentions which stress terms the DIVA approach covers. Sec. 2.2, that includes the mathematical equations of the stress balance would be suited to refer to these stress terms. I suggest to name which of the shown equations/terms correspond to which stress terms (SIA, SSA and additional stresses that are not captured by the SIA/SSA). That would give a lot more clarity on what the actual difference between DIVA and hybrid SIA/SSA is.*

Author response:
*We will add these clarifications to the text immediately after Eq. 1.*

I could not find the clarifications mentioned by the authors. If they are somewhere else, please indicate line numbers.

My point:
*16,L14: I would be interested in more details on the simplicity of the mentioned rheology, damage and subglacial hydrology. I recommend to discuss them here or to present details in the section 2.*

Author response:
*We will briefly mention the treatment of rheology and damage in section 2. The (lack of) treatment of subglacial hydrology is already included in the description of basal sliding; pore water pressure is calculated solely based on bedrock elevation, following Martin et al. (2011).*

I was not able to find the announced additions to Sec. 2 regarding the ice rheology and damage. Please indicate line numbers on where the changes were made.

Best wishes,
Johannes Feldmann

---

## Author Response (AR2)

**Rebuttal to the review by Johannes Feldmann**

We thank the reviewer for their insightful and constructive comments on our manuscript. We'd hereby like to address their concerns, and propose revisions to our manuscript to alleviate them. Reviewer's comments are displayed in boldface, replies in regular type.

**P4,Eq.(1): The meaning of the x and y indices and the bar above the variables is still not given in the text. Please add.**

We will do so.

**P6,L6-8: I suggested to remove "the square of" because otherwise from that sentenced one would deduce that Feldmann et al, 2014 and Leguy et al, 2021 also use the the square of the grounded fraction. Just wanted to point to this again but would leave it to the authors to change the statement or not.**

We agree that the current phrasing is confusing regarding the way the grounded fractions are used in PISM and CISM; we will follow the reviewer's suggestion.

**P17,L15: The authors announced but missed to include the reference Robin (1955).**

We will include this reference.

**P21,L9: investigating → investigation**

We will change this.

**I could not find the clarifications mentioned by the authors. If they are somewhere else, please indicate line numbers.**

**My point:**
**P4,Sec.2.2: The introduction mentions the advantages of the DIVA approach compared to the hybrid SIA/SSA approach and briefly mentions which stress terms the DIVA approach covers. Sec. 2.2, that includes the mathematical equations of the stress balance would be suited to refer to these stress terms. I suggest to name which of the shown**

**equations/terms correspond to which stress terms (SIA, SSA and additional stresses that are not captured by the SIA/SSA). That would give a lot more clarity on what the actual difference between DIVA and hybrid SIA/SSA is.**

**Author response:**

5 **We will add these clarifications to the text immediately after Eq. 1.**

**I could not find the clarifications mentioned by the authors. If they are somewhere else, please indicate line numbers.**

**My point:**

10 **16,L14: I would be interested in more details on the simplicity of the mentioned rheology, damage and subglacial hydrology. I recommend to discuss them here or to present details in the section 2.**

**Author response:**

**We will briefly mention the treatment of rheology and damage in section 2. The (lack of) treatment of subglacial**

15 **hydrology is already included in the description of basal sliding; pore water pressure is calculated solely based on bedrock elevation, following Martin et al. (2011).**

**I was not able to find the announced additions to Sec. 2 regarding the ice rheology and damage. Please indicate line numbers on where the changes were made.**

20

We apologise for these oversights. The changes we mentioned in the previous rebuttal were included in the manuscript, but apparently they were accidentally reverted at some point in the different versions of our document. We apologise for this oversight, and we will make sure to include them in the next revision.

25

**Rebuttal to the review by an anonymous reviewer**

We thank the reviewer for their insightful and constructive comments on our manuscript. We'd hereby like to address their concerns, and propose revisions to our manuscript to alleviate them. Reviewer's comments are displayed in boldface, replies in regular type.

We have grouped the comments about the ABUMIP experiment, the MISMIP+ experiment, and the model resolution together. Otherwise comments are addressed in order.

**1 Comments about ABUMIP**

**I thank you for adding the control experiment to the ABUMIP set of plots, it is very informative. I am concerned you thought it would be good enough to show a control that is drifting without forcing by 1.2 m of sea level rise in 500 years. At the end of that time, both Ronne and Ross ice shelves are almost gone. This is worrisome. And the linear trend is not plateauing based on your figure.**

**…**

**Before this manuscript can be published: please perform the Antarctic spin-ups at the different resolutions you are presenting that are long enough to prevent your Antarctic ice sheet to drift so much during the control experiment. Doing so would give the reader confidence you can set up your model for paleo or future Antarctic simulations. In addition, please complete your analysis by showing a difference of your end of spin-up runs with your thickness target for your run at 40km and 16km resolutions (since you plan on using both resolutions for Antarctica for paleo and future runs respectfully). Redo the ABUM and ABUK simulations starting from these new spin-ups. Redo Figure 8 accordingly.**

**…**

**P17, l14: why did you choose a spin-up of only 500 yr? Clearly you have not reached a steady state. Please continue your spin-ups until you do so. (See main comments from earlier.)**

**…**

**P18, l5: Yes, your drift in ABUC is quite large for a steady state, especially at 40km. This alone indicates that your simulation is not ready. See earlier comments.**

**…**

**P18, l7: I disagree, this should not be improved in future work and should be investigated now. This paper is a benchmark for your future scientific studies, and for this reason it is reasonable to expect that you can perform and**

**show one example of an "acceptable" spin-up. (IMAU-ICE v1.0 could do so according to their ABUC results.) All along you are claiming that you can, and it is reasonable to perform simulations at 40km. Clearly, with this result, it isn't! Arguably, it is not as well at 20km resolution. After 500 year of control experiments, both the Ross and Ronne ice shelves are already almost gone!**

**Right now, you are showing that IMAU-ICE is not ready to perform sensible Antarctic simulations. And if you cannot produce a good initial steady state for your ABUMIP experiment then you should take out the ABUMIP experiment from this paper and replace it with something else and revise your text accordingly.**

**…**

10 **Now that I have witnessed your way of performing Antarctic spin-up, I would highly encourage you to show the result of a Greenland spin-up at 20km resolution, since you plan on using this resolution for your future studies. (I am simply asking about a spin-up here, not a transient run of any kind). Such a result would strengthen the proof of concept of your new numerical capabilities (I am thinking about DIVA here).**

**…**

15 **P4, l16: in this paper you have not shown any results of any of the continental ice sheets you plan on running with except one for Antarctica (which is quite unfortunate). Showing an initial thickness from ice sheets you experimented with could add great value to your manuscript. In addition, you specifically refer to future Greenland ice sheet (GrIS) simulations here which brings me to one of my general remarks (see above) that showing a steady state simulation of the GrIS would be a great addition to this manuscript.**

20

We agree that, before our model can be used for actual future projections, we need to set up a better spin-up which has a lower model drift. The reason we chose to include the ABUMIP experiment, is that the ABUMIP experimental protocol does not require the model to be in a steady state at the start of the experiment; several of the other models in the ensemble by Sun et al. (2020) show a drift of well over half a meter of sea-level rise/fall in ABUC. Even with a model that perfectly resolves

25 the stress balance, achieving an initial state that both matches the present-day observed geometry and also has no model drift, is very likely not possible without inverting for either basal roughness and/or basal melt (which of course implies the assumption that the present-day ice sheet is indeed in a steady-state – which is very debatable!). Since we only want to benchmark the ice-dynamical component of our model in this manuscript (verification, not validation), setting up such an initialisation is beyond the scope of this publication. We will reflect these thoughts in the manuscript.

30

As a compromise, we have improved the initialisation for our ABUMIP simulations. In addition to the mechanical relaxation (now shortened from 500 yr to 100 yr), we have added a 240-kyr thermal spin-up (two complete glacial cycles) to include the "thermal memory" of the glacial maximum in the viscosity, followed by a 100-kyr mechanical relaxation with fixed shelves, to allow the grounded ice to equilibrate. This means that our spun-up geometry now deviates further from the

present-day observations than before (particularly in East Antarctica, where the ice sheet tends to "flatten out", thinning in the interior and thickening at the margins), but has negligible drift. Achieving a proper steady state in the ABUC experiment requires prescribing appropriate basal melt rates, which is admittedly difficult with the melt parameterisation described in the current manuscript. Since the basal melt formulation does not affect the ABUM and ABUK experiments, we opted instead to derive melt rates for ABUC using a simple geometry-based inversion (increasing melt rates when the shelf thickens, and vice versa; following Bernales et al., 2017). This results in a model drift of -0.1 to 0.1 m.s.l.e. after 500 years in ABUC. With this new spin-up procedure, sea-level rise in ABUM and ABUK is now smaller than in the previous set of results: 7.5 – 7.8 m for ABUM, and 7.6 – 7.9 m for ABUK. Note that, as before, the resolution dependence is still very small.

We will adapt the manuscript to reflect these changes.

One small side note: the Filchner-Ronne and Ross ice shelves did not collapse in our previous ABUC results, but they were not visible in the figure due to an unfortunate choice of colormap. The lowest bin of elevation values was shown in white, which was not visible on the white background – which escaped our notice. Thank you for pointing this out, we will fix this.

**2 Comments about MISMIP+**

**In this manuscript, you are showing some numerical capabilities about your model which is one side of the story. The other side of the story is you try to convince the reader that your model is suitable for continental scale simulations as well. (You do show the different domains of possible computation in Fig.1 after all.) So far, for this latter point, you are convincing me that your model is not ready for continental scale simulations, specifically with your ABUMIP experiments. In our first round of comment, me and another reviewer did recommend you adding some MISMIP+ experiments to show the robustness of your choice of resolution and melt parameterization especially for marine ice sheet configuration. You chose not to do so arguing that it is material for another publication. Doing so puts you at a risk that readers will not believe that your choice of model configuration for ABUMIP is suitable. It does so with me. You mentioned the length of your manuscript for not showing the MISMIP+ experiments here. I will say that this should never be the sole excuse for not adding a scientific result to a manuscript. The gmd journal is actually a good place where I can expect papers to be longer because authors are trying to show development of their models and convince the reader of their scientific capabilities (just what you are trying to do).**

**…**

**I understand you have a paper underway (hopefully) about MISMIP+ experiments to complement this one. For the time being, please add a quick highlight of your MISMIP+ experiment results supporting your default of using the FCMP parameterization from Leguy et al. (2021). (Unless you want to add these experiments to this paper to show your results at 40km down to 10km).**

We will add a paragraph about the MISMIP+ experiment. We will show the "default" simulations, which exactly follow the protocol from Asay-Davis et al. (2016). Our results for these experiments agree well with the ensemble results of Cornford et al. (2020), lying close to the ensemble mean, and well within the ensemble range. The upcoming paper about the more detailed investigation of this experiments includes simulations with a different resolution, stress balance, sliding law, sub-grid melt scheme, and basal melt parameterisation. This new paper is almost finished, and we expect to submit this within a few weeks.

**3 Comments about model resolution**

**I feel a bit uneasy with a current tendency of ice sheet modeling papers stating it is fine to use coarse resolution (here 40km) and allow the model to have greater error because it will be used for paleoclimate simulations. I respectfully disagree with this mindset, and the computational expense of long paleoclimate simulations should not be an excuse to (gravely) misrepresent ice sheet behavior. In this paper you also mention that you plan on using the model for future projection study as well (using 16km resolution) meaning this "benchmark" paper should also be used to convince the reader it is acceptable to do so with this model, which it isn't.**

**In my first round of review, I did request for the sections presenting the experiments to be a bit more quantitative in their discussions; sometimes adding tables is also a good idea. I appreciate an effort was made in adding convergence figures (e.g., figure 7b) and I would like to see more discussion on the relative errors whether with respect to the analytical solutions or with the highest resolution used in these experiments. Simply stating something like "look, we do see convergence with resolution" is over simplistic. In Sec3.2, Fig.3, it is not only difficult to get a sense of the error with respect to the analytical solution (the y axis is not precise enough) but also you state (in your discussion) that your error at 40km is a reliable result for paleoclimate studies. Two arguments here: 1. What is an acceptable/reliable error? You never define this concept (and it is hard to do actually). 2. You seem to argue that an error of 350m/y with respect to the analytical solution is acceptable. I would beg to differ! And this is where a more detailed discussion either here or in the discussion needs to take place explaining why this magnitude of error is good enough for paleoclimate simulations. Based on the results on this section, I would feel way more comfortable if you chose to use a resolution of 20km.**

**…**

**P17, l6: This argument alone is insufficient to justify an acceptable result. (If I run an ice sheet model with a resolution of 200km and see an error of 195km, does it make it acceptable to use such a resolution?) Also, this part of your sentence is a repetition of what you already said 2 sentences prior. At 40km, the hysteresis corresponds to about 25% of the grounding line displacement. Here you are benchmarking your model meaning you will refer to it to explain results of**

**future science experiments. I would advise to acknowledge this large error, and in the discussion, give an example of a situation for coupled climate models for which this error could be of small importance.**

**Deeming an error to be acceptable is a tricky business and at the end modelers will run with whatever they feel comfortable with to justify their science.**

5 **What I can see in these experiments is that, again, running with a resolution of 20km leads to well improved results compared to 40 with again a superlinear convergence. Any idea what is happening with your 16km resolution results? This might be of importance since you plan on using this resolution to run continental scale future scenarios. Based on this experiment, it seems that 20km is better suited.**

**…**

10 **P19, l32: I strongly disagree with this statement, see previous comments. Define "reliable results"? This section is perhaps a good place to spend time on arguing why you think this resolution is adequate and cite references to back up your statements. Just saying it does not make it true.**

It is important to keep in mind that palaeo-ice-sheet simulations generally are interested in large-scale ice-sheet
15 evolution. When studying glacial cycles, we don't want to know at exactly what rate the grounding line of one particular outlet glacier is retreating during a century; we want to know how many tens of metres of sea-level change we can expect after ten thousand years. This is not just because that is the kind of quantity we can actually compare to proxy data, but also because the uncertainties in the forcing (palaeoclimate, paleogeography) are so large. Comparing modelled small-scale ice-sheet features to data becomes meaningless when the large-scale forcing is so uncertain, and so we generally tolerate such small
20 features becoming obscured by a low resolution. In all of our experiments that concern large-scale dynamic evolution (Halfar dome, Bueler dome, MISMIP, ABUMIP), our model produces robust results even at 40 km resolution. The only experiment where we do indeed see some resolution dependence is the SSA ice-stream experiment mentioned by the reviewer, which does not concern dynamic evolution, but only instantaneous velocities. The most realistic and representative experiment is ABUMIP, and there we see no appreciable difference between the different resolutions (not in the previous set of results, and
25 not in the new). This gives us confidence that, even at 40 km resolution, our model can simulate large-scale ice-sheet geometry and (rates of) sea-level change accurately enough that the model errors are negligibly small compared to the forcing errors typical of palaeo simulations. We will clarify these thoughts in the manuscript.

Regarding the reviewer's statement about computational expense: the harsh reality is that computational resources
30 are limited, no matter how much we'd like them not to be. Every doubling of the resolution increases the computation time by about a factor 10. For a few one-off experiments that could be achievable, but palaeo-modelling usually involves sensitivity analysis based on ensemble simulations. We therefore partly agree with the reviewer that we should aim to always include a few higher-resolution runs in our ensemble studies, but it is not something we can feasibly do for every experiment (especially since the importance of accounting for the long paleo histories of the Greenland and Antarctic ice sheets in projections of their

future retreat is becoming increasingly clear, e.g. Yang et al., 2022: Impact of paleoclimate on present and future evolution of the Greenland Ice Sheet, PLoS ONE 17, https://doi.org/10.1371/journal. pone.0259816). We do not believe that means we should not do any palaeo modelling at all until we come up with a faster model or a bigger computer; it just means that we have to accept that our results are not as accurate as they might at some future time become. We will reflect these thoughts in

5 the manuscript.

**4 Other comments**

**P7, l5: I suggest rewriting "The way the stress balance is discretize" by "The stress balance discretization".**

We will do so.

**P7, eq10 and 12: my previous comment on these equations might have been misunderstood. I appreciate you adding the bounds 0<=w_b<=1 and 0<=\lambda_w<=1 within the equations for clarity but it is still confusing. What I would**

15 **like to see specifically written in the text is:**

**1. that for equation 10, b is bounded between b_min and b_max and you achieve this by writing something like:**

**w_b = ! ! max$(0, b − b\%\&', )$, b < 0 "!"##"_%&' min$(b\%() − b\%\&', b − b\%\&', )$, b > 0**

**2. that d_w is bounded between 0 and 1000 so that the bounds for d_w are satisfied. Right now, you write d_w = z_SL -b. So if z_SL=0 (say see level reference set to today's value) and b=- 1100, then d_w=1100 and lambda_w=1.1. So**

20 **clearly, you are taking the minimum between 1 and 1.1 here. Conversely, if b is above sea level, d_w<0 and in this case, you are taking the maximum between d_w and 0. So please, for clarity write d_w using something like:**

**d_w=!max$(b\%(),z*+ −b)$,b>0 min$(−b\%\&',z*+ −b)$,b<0**

**3. After defining your equations, in the text (and you deleted it), mention that both w_b and d_w are capped between 0 and 1.**

25

The scaling coefficients w_b and lambda_w are limited between 0 and 1 after they are calculated, e.g. if d_w = 1100, then lambda_w = 1. We will clarify this in the text and adjust the equations accordingly.

**P10, Table 3: None of the parameters listed in this table are defined in the text. Please do so either in the table and/or**

30 **in the text.**

We will add a "description" column to Table 3, similar to Table 2.

**P10, l13: you deleted the sentence stating the possible melt parameterization options available in IMAU-ICE v2.0. I believe it is good to keep them in the text and stating the default configuration for your version of the model. In the previous version you did mention these options but did not discuss the effect of them on your model output. If you do have results comparing them in the context of Antarctic simulation, please, add this discussion!**

We will restore the description of the optional PMP and NMP sub-grid melt schemes, and mention the effects these have on the results of the MISMIP+ and ABUMIP experiments in some preliminary tests.

**P12, l3: Can you be a bit more quantitative? Also, the convergence with resolution seems to be superlinear (at least at the center of the domain) and there is a net gain in using a resolution of 20km as opposed to 40km in these simulations. Why do you think the error at 40km resolution is acceptable? (See previous comments.) (I believe the new figure 3 is more informative compared to the older one; we learn something more about the model itself. Thank you for redoing it.)**

We will add a panel to the figure showing the convergence, similar to the ones already included for the Halfar & Bueler domes and the MISMIP experiments. The order of convergence is about 1.7, which is indeed more than linear, and close to the value of 1.9 reported by Bueler and Brown (2009).

As mentioned before, we do not believe this experiment is very informative about the performance of the model in simulating the dynamic evolution of an ice sheet, as it only concerns instantaneous velocities in a confined setting. Our conclusion that 40 km is satisfactory is supported by the other experiments we present.

**P12, Table 4: please, align your table headers (Parameter, description,...) on one line.**

We will fix this.

**P16, l8: Please rephrase this sentence. The ice flow factor is decreased (increased) as a step function after which you run your model forward in time for 15 kyr (not 25 kyr based on Fig. 7A) to a steady state. Please, add the values for the ice flow factors in your text.**

We will do so.

**P18, Fig.8: The caption indicates that the results from IMAU-ICE v1.0 are shown by dashed black line. I can only see a plain line. Also, which resolution are you displaying on the left? (And thank you for redoing this figure, it is much easier to read and more informative compared to the previous version.)**

The results om IMAU-ICE v1.0 are shown by a thick grey line; we will correct the caption. The panels on the left show the 10 km results; we will mention this in the caption.

**P19, l4: Why should this be as expected? I don't think so. Here you are comparing your results with ELMER/ice for the ABUM experiment. In the next sentence you argue that the difference between ABUM and ABUK should be small for a given model within 3-4 years of transient simulation (I'll get back to this later). The ELMER/ice result for ABUK is almost 8m lower than the one of ABUM after 500 years. Why do you think there ABUM results is correct and their ABUK result is wrong? I would argue the opposite!**
**I am not saying your results are wrong per se, I am saying that your argument for comparison is not sound. Your model configurations are different compared to all the models in Sun et al. (2020). And this is where your melt parameterization analysis could help you out.**

We did not mean to argue that our results were correct because they were similar to ELMER; we will remove the reference to ELMER in the text. What we do mean to argue is that, purely from the physics, we do not expect to see much of a difference between the ABUK and ABUM experiments (although we do acknowledge that there is some room for discussion; we will phrase it less strongly in the manuscript). The fact that some other models in the Sun et al. ensemble, including ELMER, show substantial differences is certainly interesting, but explaining why this is the case is beyond the scope of our work. The fact that our own model does not show much of a difference is, in our view, supportive of our model's performance.

**P19, l27: Some of your results are close to analytical solutions but they do not match! Also, you only compared to ELMER/ice and otherwise your ABUMIP results are outliers compared to other ice sheet models. Please rephrase**

We will find another phrase to indicate that our results closely approximate the analytical solutions.

**P20, l6: replace "With a minimum of effort" by "With minimum effort".**
We will do so.

**P20, l7: a space is missing after ")"**
There is not; parentheses around an optional prefix should not have a space between the closing parenthesis and the word being prefixed.